# WATERMARKING DIFFUSION LANGUAGE MODELS

**Thibaud Gloaguen, Robin Staab, Nikola Jovanović, Martin Vechev**
ETH Zurich
thibaud.gloaguen@inf.ethz.ch

## ABSTRACT

We introduce the first watermark tailored for diffusion language models (DLMs), an emergent LLM paradigm able to generate tokens in arbitrary order, in contrast to standard autoregressive language models (ARLMs) which generate tokens sequentially. While there has been much work in ARLM watermarking, a key challenge when attempting to apply these schemes directly to the DLM setting is that they rely on previously generated tokens, which are not always available with DLM generation. In this work we address this challenge by: (i) applying the watermark in expectation over the context even when some context tokens are yet to be determined, and (ii) promoting tokens which increase the watermark strength when used as context for other tokens. This is accomplished while keeping the watermark detector unchanged. Our experimental evaluation demonstrates that the DLM watermark leads to a >99% true positive rate with minimal quality impact and achieves similar robustness to existing ARLM watermarks, enabling for the first time reliable DLM watermarking. Our code is available here.

## 1 INTRODUCTION

While autoregressive language models (ARLMs) have demonstrated remarkable performance (OpenAI et al., 2024; Grattafiori et al., 2024; Bubeck et al., 2023), other approaches to language modeling, such as diffusion language models (DLMs), are rapidly catching up, with recent works (Nie et al., 2025; Ye et al., 2025; Labs et al., 2025) approaching similar capabilities at a significantly lower cost. Notably, DLMs have higher generation speed, offer built-in error correction, are more controllable, and can be naturally extended to multiple modalities (Yang et al., 2025). These advances also increase the risk of misuse, making it paramount to reliably detect text generated by these models. Prior works (Kirchenbauer et al., 2023; Kuditipudi et al., 2024; Christ et al., 2024) have proposed the concept of ARLM watermarks: the generated text is augmented with an imperceptible signal that can later be detected to trace its provenance. Such watermarks are already adopted in consumer-facing models (Dathathri et al., 2024) and are being advocated for through regulation (EU Council, 2024). Yet, most existing works on language model watermarking rely on hashing mechanisms compatible only with autoregressive generation, highly limiting their application for DLMs. Our work is the first to attempt to overcome this limitation, enabling efficient and reliable watermarking for DLMs.

**ARLM Watermarks** Watermarks for ARLMs traditionally rely on three key components. The *hashing mechanism* (i) uses the previously generated tokens to seed the *sampling procedure* (ii), which then inserts the watermark signal into the generated text. Critically, this pipeline relies on the previous tokens to have already been generated to compute the hash, an assumption often violated with DLMs. The *watermark detector* (iii) leverages the hashing mechanism to retrieve the seed associated with each token and compute a test statistic to determine whether a given text is watermarked.

**Diffusion Language Models** DLMs produce a probability distribution over the set of fixed-length sequences containing masked tokens, which, analogous to noise in continuous diffusion models, represent placeholders yet to be generated. Starting from a (partially) masked sequence, DLMs iteratively sample (*unmask*) tokens until the sequence is fully generated. Importantly, unlike autoregressive models, DLMs are not constrained to unmasking tokens in a left-to-right order, making hashing-based ARLM watermarks inapplicable whenever a token without full prior context is unmasked. A natural workaround is to restrict the watermark application to tokens with fully available context. Yet, as we show in Sec. 4, such a naive approach leads to a weak watermark, as barely any tokens fulfill this criterion. This motivates the need for a novel watermarking algorithm tailored to DLMs.

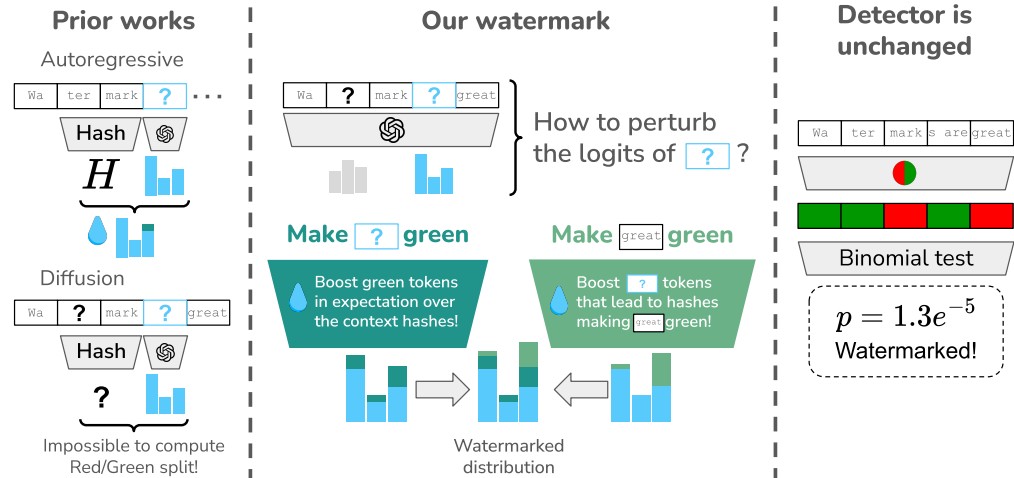

Figure 1: An overview of why current watermarks for ARLMs fall short in the diffusion setting (left), how our watermark operates in this setting (middle) and how our watermark detector works (right).

**This Work: Watermarking Diffusion LMs**    In this work, we introduce the first watermarking scheme tailored to DLMs based on the prominent Red-Green watermarks (see Sec. 2), as illustrated in Fig. 1. First, we identify the limitations of autoregressive hashing in most prior works on ARLM watermarks when applied in the diffusion setting (Fig. 1, left). To overcome this limitation, we frame watermarking as a constrained optimization problem (Sec. 3.1). From this optimization problem, we then derive a practical watermarking scheme for DLMs (Sec. 3.2). In Sec. 3.3, we interpret our scheme as a combination of two intuitive components: applying ARLM watermarks in expectation over the context hashes and biasing tokens that lead to hashes making other tokens green (Fig. 1, middle). Our detector is exactly the same as in prior Red-Green watermarks: we compute the color of each token in the sequence and perform a binomial test (Fig. 1 (right)). In our experiments, we show that our watermarking scheme is significantly more detectable than naive baseline adaptations of ARLM watermarks (exceeding 99% TPR at 1% FPR with negligible quality degradation), and is robust to common modifications of the generated text, e.g., substituting words in context (Sec. 4).

**Main Contributions**    Our key contributions are:

- We conduct the first study of watermarking for DLMs, identifying the limitation of ARLM watermarks in the diffusion setting and proposing a watermarking scheme tailored for DLMs.

- We formulate the problem of DLM watermarking as a constrained optimization problem (Sec. 3.1) from which we derive a practical watermarking scheme that leverages existing watermark detectors from prior works (Sec. 3.2).

- We further interpret our watermarking scheme as an extension of existing ARLM watermarks (Sec. 3.3) and demonstrate that, when restricted to the autoregressive case, our optimization formulation recovers exactly ARLM watermarks (Sec. 3.3).

- Through extensive evaluation, we show that our scheme is suitable for practical use, preserves the generated text quality and is robust against common natural text modifications (Sec. 4).

## 2    BACKGROUND AND RELATED WORK

**Language Modeling**    The goal of language modeling (Dai & Le, 2015; Devlin et al., 2019) is to train a model $\theta$ to learn a probability distribution $p_\theta$ over natural language. In particular, given a vocabulary $\Sigma$ and the true probability distribution $p_{\text{data}}$, the goal is to find $\theta$ such that $p_\theta \approx p_{\text{data}}$. With autoregressive modeling, given a text $\omega \in \Sigma^*$, its probability is factorized sequentially using the (probability) chain rule. While this formulation allows for efficient training (Radford et al., 2018; 2019; Brown et al., 2020), it forces ARLMs to generate new tokens sequentially, which fundamentally limits their generation speed and might restrain their capabilities (Berglund et al., 2023). Recent works (Gloeckle et al., 2024) have proposed multi-tokens prediction as a training objective to improve model capabilities, but crucially still generate tokens autoregressively.

**Diffusion Language Modeling** In contrast, in diffusion language modeling (Austin et al., 2021a; Lou & Ermon, 2023; Shi et al., 2024; Ou et al., 2024; Sahoo et al., 2024), the vocabulary $\Sigma$ is extended with a mask token representing a position where a token is yet to be sampled. Let $\tilde{\Sigma}$ be the vocabulary *including the mask token*. The diffusion process is parameterized by a fixed number of diffusion steps $N$ and a generation length $L$. At each diffusion step $i$, the model takes a sequence of tokens $\tilde{\omega}^{(i)} \in (\tilde{\Sigma})^L$ and returns a *factorized* probability distribution $p^{(i)} \in \Delta(\Sigma)^L$ over the set of sequences, where $\Delta$ denotes the simplex. Each token in the sequence of length $L$ is then sampled independently from $p^{(i)}$, which returns an intermediary sequence $\omega^{(i+1)} \in \Sigma^L$. This sequence is then masked $\tilde{\omega}^{(i+1)} := Mask(\omega^{(i+1)}, p^{(i)})$, using a pre-determined stochastic masking procedure $Mask \colon \Sigma^L \times \Delta(\Sigma)^L \to (\tilde{\Sigma})^L$. We say a token is getting unmasked if it was masked in $\tilde{\omega}^{(i)}$ but not in $\tilde{\omega}^{(i+1)}$. This process is iterated $N$ times, returning a final $\omega^{(N+1)} \in \Sigma^L$ which has no mask tokens. Unlike ARLMs, DLMs can, in each step, generate (unmask) multiple tokens and in any order. We evaluate our watermark with multiple unmasking procedure $Mask$: uniform (Austin et al., 2021a), based on distribution entropy (Kim et al., 2025), and by block (Arriola et al., 2025).

**Large Language Diffusion Models** With the release of open Large Language Diffusion Models (Nie et al., 2025; Ye et al., 2025), DLMs have gained significant traction in multiple domains: reasoning (Sahoo et al., 2025; Cetin et al., 2025), chemistry (Tan et al., 2025), multimodality (Yang et al., 2025), vision (You et al., 2025) and safety (Ma et al., 2024). For traceability, previous works (İsmail Tarım & Onan, 2025) only explored zero-shot detection of DLMs generated text, and highlighted the lack of a reliable watermarking solution in this paradigm.

**LLM Watermarks** The goal of generative AI watermarks is to ensure content traceability, i.e., the ability to rigorously identify whether a given piece of content has been generated by an AI model. Key challenges for watermarks include balancing their impact on content quality, their strength, and their robustness to edits (Tu et al., 2024; Pan et al., 2024), while ensuring their reliability against false positives. For ARLMs, a range of *generation-time* watermarks (Kirchenbauer et al., 2023; Kuditipudi et al., 2024; Aaronson, 2023; Dathathri et al., 2024; Christ et al., 2024) have been proposed by building on the autoregressive nature of the underlying models. Prior works have also proposed order-agnostic watermarks, namely Unigram (Zhao et al., 2023) and its extension PatternMark (Chen et al., 2025), that can be directly applied to DLMs. However, such approaches significantly compromise on watermark security (Jovanović et al., 2024; Zhang et al., 2024). We nonetheless show in Sec. 4.4 that our approach, tailored for DLMs, outperforms such schemes. Other concerns regarding watermark design include their applicability in the open-source setting (Gloaguen et al., 2025; Xu et al., 2025).

In this work, we study the most popular family of current ARLM watermarks: *Red-Green watermarks* Kirchenbauer et al. (2023). Let $\omega_t \in \Sigma$ denote the token generated by the LM at step $t$, and $k$ the *context size* parameter. Using a hash of the context $H(\omega_{t-k:t})$, a pseudo-random function partitions the vocabulary $\Sigma$ into a *green* subset and the remaining *red* subset. The size of the green subset is set to $\gamma|\Sigma|$, with $\gamma \in (0,1)$ commonly chosen as $0.25$ or $0.5$. To insert the watermark, each green token in the logits vector gets shifted up by a constant $\delta$, increasing the overall likelihood of sampling a green token. To detect the watermark, given a sequence of tokens $\omega \in \Sigma^*$, we extract a corresponding color sequence and perform a *binomial test* on the number of green tokens.

**Image Diffusion Watermarking** With the increasing popularity of diffusion models for image generation (Dhariwal & Nichol, 2021; Ho et al., 2020; Nichol & Dhariwal, 2021; Song et al., 2020), image diffusion watermarking has been the focus of many works (Fernandez et al., 2023b; Wen et al., 2023; Yang et al., 2024). Yet, all generation-time image diffusion watermarks operate in a continuous space, making them fundamentally inapplicable to the discrete diffusion process of DLMs.

## 3 WATERMARKING DIFFUSION LANGUAGE MODEL GENERATION

In this section, we present our approach to watermarking DLMs. Sec. 3.1 introduces a theoretical framework to guide our DLM watermark design, while Sec. 3.2 addresses practical challenges and proposes an instantiation of our watermark. Lastly, in Sec. 3.3, we see how our watermark naturally extends Red-Green ARLM watermarks: it decomposes into the two terms illustrated in Fig. 1 (middle) and, when restricted to the ARLM case, it instantiates to Red-Green ARLM watermarks.

### 3.1 ADAPTING WATERMARKS TO THE DIFFUSION SETTING

We propose framing the LM watermarking algorithm as an optimization problem, which in turn allows us to adapt watermarks for autoregressive language models to the diffusion setting.

**Naive Approach** As alluded to in Sec. 1, a naive approach to adapting Red-Green watermarks in the diffusion setting would be to apply them only to tokens whose context is fully unmasked. Indeed, for such token positions, computing their context hash and thus the watermark distribution is possible. However, this means that we cannot apply the watermark (e.g., boost the probability of green tokens) to any token that is unmasked before its respective context. Ultimately, this would result in only a handful of green tokens which, as we show in Sec. 4.1, leads to a low true positive detection rate. Importantly, we find that a watermark tailored for DLMs needs to operate directly over the *distribution* of context hashes, enabling it to watermark all tokens regardless of the unmasking order.

**Goal of Red-Green Watermarks** Given a sequence $\omega$, the detector computes the proportion of green tokens $\hat{\gamma}(\omega)$ within that sequence and conducts a binomial test. If $\hat{\gamma}(\omega)$ significantly exceeds the baseline ratio of green tokens expected under the null hypothesis $\gamma$, the sequence $\omega$ is identified as watermarked. Thus, the goal of the watermarking algorithm is to modify the sampling procedure such that each generated sequence $\omega$ has a high ratio $\hat{\gamma}(\omega)$, while preserving the model's utility.

**Watermarking as a Constrained Optimization Problem** A watermarked DLM should preferentially sample token sequences that maximize the ratio $\hat{\gamma}(\omega)$. This naturally leads to framing the problem of DLM watermarking as a constrained optimization task: we aim to maximize the expectation of the green token ratio over the generated sequence while minimizing the impact on model performance. We show in Sec. 3.2 that our proposed optimization framework abstracts the complexity of DLM watermarks into a practical watermarking algorithm, illustrated in Fig. 1 (middle).

Recall that $\Sigma$ is the model vocabulary, and $\tilde{\Sigma}$ is the vocabulary extended with the mask token. Let $\omega \in \Sigma^L$ be an unmasked sequence of tokens. For every token position $t$, we introduce a hash function $H_t : \Sigma^L \to \mathcal{H}$ that returns the hash of the context at position $t$, with $\mathcal{H}$ the *finite* set of possible hash values. We assume that the token $t$ does not belong in its own context (no self-hashing assumption). We further discuss this assumption in App. H. We introduce the *global* binary green list matrix $G \in \{0,1\}^{\mathcal{H} \times \Sigma}$ which, given the context hash and a token, returns its color. We formalize the *green ratio function* $\hat{\gamma} : \Sigma^L \to [0,1]$ as $\hat{\gamma}(\omega) = \frac{1}{L} \sum_{t=1}^{L} G_{H_t(\omega), \omega_t}$, i.e., the color $G_{H_t(\omega), \omega_t}$ of each token $\omega_t$ averaged over the sequence $\omega$.

Let $\tilde{\omega} \in (\tilde{\Sigma})^L$ be a (partially) masked sequence and $p(\tilde{\omega}) \in \Delta(\Sigma)^L$ the corresponding *factorized* probability distribution over the set of sequences returned by the DLM forward pass. Given $t \in [1, \dots, L]$, $p_t$ is the probability over $\Sigma$ at position $t$. The goal of the watermarking algorithm is to distort the factorized probability distribution $p(\tilde{\omega})$ into a factorized probability distribution $q(\tilde{\omega})$ that maximizes the expected green ratio of sequences sampled according to $q(\tilde{\omega})$. We have to solve,

$$q^* = \underset{q \in \Delta(\Sigma)^L}{\arg\max} \, \mathbb{E}_{\Omega \sim q}[\hat{\gamma}(\Omega)], \text{ subject to } \forall t \in [1, \dots, L], \mathrm{KL}(q_t, p_t(\tilde{\omega})) \leq \varepsilon. \quad (1)$$

In Eq. (1), the KL constraint with $\varepsilon > 0$ is used as a proxy for controlling impact on quality. We now explain how to exactly compute the expectation from Eq. (1). For any $q \in \Delta(\Sigma)^L$, to compute the expectation, given $\Omega \sim q$ (i.e., $\Omega$ is the random variable representing sequences distributed according to the factorized probability distribution $q$), we need to know the distribution of $\Omega_t$ and $H_t(\Omega)$. The distribution of $\Omega_t$ is by definition $q_t$. Computing the distribution of $H_t(\Omega)$, i.e., the distribution of the context hashes, is more challenging. We model it for every token position $t$ as a function of $q$, $h_t : \Delta(\Sigma)^L \to \Delta(\mathcal{H})$, and defer instantiations to Sec. 3.2. We can now unroll the expectation,

$$\forall q \in \Delta(\Sigma)^L, \mathbb{E}_{\Omega \sim q}[\hat{\gamma}(\Omega)] = \frac{1}{L} \sum_{t=1}^{L} \mathbb{E}_{\Omega \sim q}[G_{H_t(\Omega), \Omega_t}] = \frac{1}{L} \sum_{t=1}^{L} h_t(q)^\top \cdot G \cdot q_t =: \frac{1}{L} J(q), \quad (2)$$

where the factorization comes from the no self-hashing assumption (i.e., $H_t(\Omega) \perp \Omega_t$). When expanding the dot products, each element is the color of a (token, context hash) tuple weighted by its probability. For clarity, we refer to $J$ as the *energy* function. The watermark is equivalent to

$$q^* = \underset{q \in \Delta(\Sigma)^L}{\arg\max} \, J(q), \text{ subject to } \forall t \in [1, \dots, L], \mathrm{KL}(q_t, p_t(\tilde{\omega})) \leq \varepsilon. \quad (3)$$

Solving Eq. (3) is challenging due to the KL constraint. Without the constraint, a greedy algorithm would suffice. In Theorem 3.1, we provide an implicit solution to Eq. (3), with the proof in App. J.

**Theorem 3.1.** *Given $p \in \Delta(\Sigma)^L$ and $J$ defined in Eq. (2), there exists $\delta \in \mathbb{R}^L$ such that*

$$\forall t \in [1, \ldots, L], q_t^* \propto p_t \exp(\delta_t \alpha_t(q^*)) \tag{4}$$

*with $\alpha_t(q) = \nabla_{q_t} J(q)$. Moreover, for all $t \in [1, \ldots, L]$, $\delta_t$ is the unique solution to $KL(q_t^*, p_t) = \varepsilon$.*

This result means that the distribution $q^*$ is optimal with respect to our optimization problem, i.e., it is the optimal way to turn any distribution $p$ provided by our DLM into a distribution that maximizes the expected green ratio while enforcing the KL constraint. Importantly, it has a fairly simple form. If we take Eq. (4) in the logits space, we see that the optimal solution corresponds to adding $\delta_t \alpha_t(q^*)$ to the logits vector—a similar distortion to the Red-Green ARLM watermark. In Sec. 3.3, we provide an intuitive explanation to how our watermark works. We show that $\delta_t \alpha_t(q^*)$ can be decomposed into two components: one that makes the token at position $t$ green and one that makes tokens which have the token at $t$ in their context green, as illustrated in Fig. 1 (middle).

## 3.2 Our Watermark Algorithm for Diffusion LMs

In this section, we instantiate the theoretical derivation from Sec. 3.1 to derive a practical watermark algorithm. First, we explain how to solve Eq. (4) and find $\delta$. We then detail instantiations of common hash functions and propose a corresponding practical watermark algorithm.

**Solving the Functional Equation**   Eq. (4) defines a functional equation with $q^*$ and $\delta$ as the unknowns. We propose using a fixed-point iterative approach of $f \colon q \mapsto p \exp(\delta \alpha(q))/Z(q)$, which gives $q^* = f(q^*)$. At each step $i$, given a $q^i$, we first find $\delta$ for this $q^i$ and then compute $q^{(i+1)} = f(q^{(i)})$. We find in Sec. 4.3 that in practice a single iteration already yields a strong watermark, and that, despite the lack of theoretical guarantees of convergence to $q^*$, increasing the number of iterations indeed slightly improves watermark strength.

To find $\delta$, we can solve for all $t$ the equation $KL(q_t^*, p_t) = \varepsilon$ using bisection, as Theorem 3.1 guarantees the existence and uniqueness of $\delta$. A simpler relaxation of our framework, closer to the parameterization of Red-Green ARLM watermarks, is to directly parameterize the optimization problem, and thus the watermark algorithm, by a constant $\delta \in \mathbb{R}$ instead of $\varepsilon$. We refer to using $\varepsilon$ as *$\varepsilon$-parameterization* and using $\delta$ as *$\delta$-parameterization*. We compare both approaches in Sec. 4.3.

**Instantiating the Hash**   We now instantiate the hash $H$ and the hash distribution $h$ from Sec. 3.1. As illustrated in Algorithm 1, the hash is used to seed a pseudo-random function to generate a row of $G$. Without loss of generality, we further assume that $\Sigma = \{1, \ldots, |\Sigma|\}$ and

---

**Algorithm 1** Green matrix $G$ row generation

---
**Require:** A hash $s \in \mathcal{H}$, watermark key $\xi$
1: $G_{s,:} \leftarrow \text{PseudoRandomGenerator}(s, \xi)$
2: **return** $G_{s,:}$

---

$\mathcal{H} = \{1, \ldots, |\mathcal{H}|\}$. The exact cardinality of $\mathcal{H}$ depends on the specific hash implementation. For the hash we consider in this work, it is systematically on the order of $|\Sigma|$. Similarly to Kirchenbauer et al. (2024), we explore two local hash functions, *SumHash* and *MinHash* (we provide a detailed comparison with the formulation of Kirchenbauer et al. (2024) in App. G.1). However, unlike in Kirchenbauer et al. (2024), for Eq. (2) to hold, we must respect the no self-hashing assumption. As alluded to in Sec. 3.1, computing $h$ naively is challenging and requires $O(\Sigma^L)$ operations. Thanks to the specific local structure of the considered hash functions detailed below, we show that the corresponding hash distributions can be computed efficiently through algebraic manipulation.

SumHash sums the surrounding tokens. Let $k \in \mathbb{N}$ be the context size, and $\mathcal{C} = \{c_1, \ldots, c_k\}$ the set of positions that define the context, where $\forall i, c_i \neq 0$. For instance, $\mathcal{C} = \{-1, 1\}$ means that, for every token position, the context is comprised of the previous token and the next token. In contrast to the ARLM setting, the context is no longer restricted to preceding tokens but can also include tokens after $t$. We define (with $\omega_t$ the token at position $t$ in the sequence $\omega$),

$$\forall \omega \in \Sigma^L, \forall t \in [1, \ldots, L], H^{SumHash}(\omega)_t = \sum_{i \in \mathcal{C}} \omega_{t+i}. \tag{5}$$

We can derive an analytical formula for the probability distribution over the hashes $h^{SumHash}$ as

$$\forall p \in \Delta(\Sigma)^L, h_t^{SumHash}(p)_s = \sum_{\substack{u_1,\ldots,u_k \in \Sigma^k \\ H^{SumHash}(u_1,\ldots,u_k)=s}} \prod_{j=1}^k p_{t+c_j}(u_j) = (p_{t+c_1} * \ldots * p_{t+c_k})_s, \tag{6}$$

where $*$ is the convolution product and $s \in \mathcal{H}$. Using the Fast Fourier Transform to compute the convolution product, computing $h^{SumHash}$ takes $O(|\mathcal{C}||\Sigma|\log|\Sigma|)$ operations.

With MinHash, the hash corresponds to the token id of the minimum of the surrounding tokens, i.e.,

$$\forall \omega \in \Sigma^L, \forall t \in [1,\ldots,L], H^{MinHash}(\omega)_t = \min_{i \in \mathcal{C}} \sigma(\omega_{t+i}), \tag{7}$$

with $\sigma$ a random permutation, to ensure randomness in the token id ordering. For $t \in [1,\ldots,L]$, $u \in \Sigma$, let $p_t^\sigma(u) = p_t(\sigma^{-1}(u))$. We have for $s \in \mathcal{H}$ and $p \in \Delta(\Sigma)^L$,

$$h_t^{MinHash}(p)_s = \sum_{\substack{u_1,\ldots,u_k \in \Sigma^k \\ H^{MinHash}(u_1,\ldots,u_k)=s}} p_{t+c_1}(u_1)\ldots p_{t+c_k}(u_k) = A_t(s+1) - A_t(s), \tag{8}$$

where $A_t(s) := \prod_{i \in \mathcal{C}} \sum_{u=s}^{|\Sigma|} p_{t+i}^\sigma(u)$. Hence, computing $h^{MinHash}$ requires $O(|\mathcal{C}||\Sigma|)$ operations.

**Our Watermark**  Based on the above methods, we now present our watermarking algorithm (Algorithm 2). Specifically, *at each diffusion step $i$*, given the distribution of the DLM $p(\tilde{\omega}^{(i)})$, we first compute the hash distribution $h$ (line 3) using Eq. (6) or Eq. (8) and subsequently derive the corresponding energy function $J$ (line 4) as per Eq. (2). For practical reasons, for each token position $t$, we only use the top-k $h_t$ and $p_t$ (we ablate over practical choices of $k$ in App. C). For each token position, we then calculate the gradient of the energy function with respect to the token's probability distribution (line 6), and

---

**Algorithm 2** Watermark for DLMs (simplified)

**Require:** DLM Probabilities $p$, number of iterations $n$, watermark strength $\delta$, generated sequence length $L$.
1: $h \leftarrow \text{HashProbabilities}(p)$
2: **for** $t$ from 1 to $L$ **do**
3:     $\alpha_t \leftarrow \nabla_{p_t} J$
4:     $q \leftarrow p_t \exp(\alpha_t \delta)$
5:     $q \leftarrow \text{Normalize}(q)$
6: **end for**
7: **return** $q$

---

apply an exponential tilt proportional to this gradient scaled by the strength parameter $\delta$ (line 7), resulting in the final watermarked distribution (line 8), as per Eq. (4). The distribution is further refined by iterating the tilting procedure. For detection, we use the same detector as Red-Green ARLM watermarks: given $\omega$, we compute each token's color and perform a binomial test on the number of green tokens after deduplication of (context hash, token) pairs (Fernandez et al., 2023a). Algorithm 2 time complexity, with SumHash, is $O(nL|\mathcal{C}||\Sigma|\log|\Sigma|)$. Yet $n, |\mathcal{C}|$ are small which results in minimal generation overhead (App. I.1).

### 3.3 INTERPRETING OUR DIFFUSION LM WATERMARK

We now provide an intuitive explanation on how our watermark distorts the original model probability distribution $p$ into a watermarked probability distribution $q^*$. We perform an explicit computation with SumHash to interpret the components of our watermark and extend this interpretation in App. J.2 for any hash function and context. Additionally, in App. E, we show that instantiating our optimization problem (Eq. (1)) for the ARLM case preserves only the boost component and corresponds exactly to the Red-Green ARLM watermark.

**Interpreting Our Watermark**  We analyze our watermarking algorithm with SumHash and $\mathcal{C} = \{-1\}$. In this setting, the energy function simplifies to

$$J(p) = \sum_{t=1}^L p_{t-1}^T \cdot G \cdot p_t, \tag{9}$$

which in turn yields, using a fixed $\delta$ and at the first step of the fixed-point iteration,

$$\forall t \in [1,\ldots,L], q_t^* \propto p_t \underbrace{\exp(\delta G^\top p_{t-1})}_{\text{expectation boost}} \underbrace{\exp(\delta G p_{t+1})}_{\text{predictive bias}}. \tag{10}$$

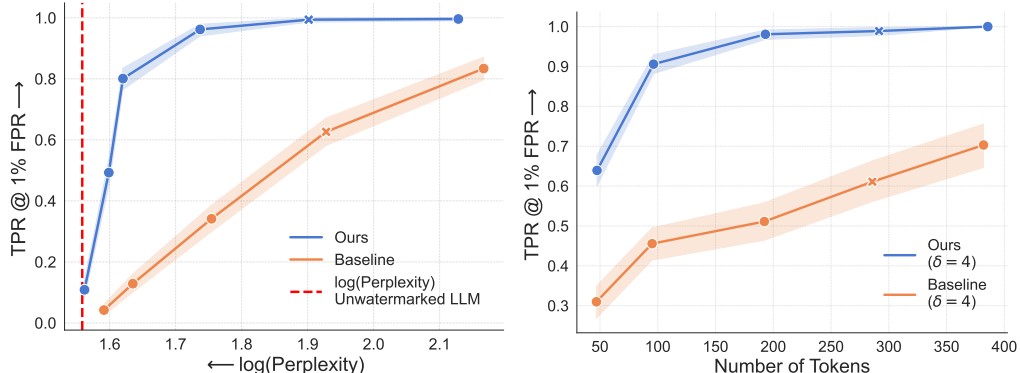

Figure 2: **Detection Performance of Our Approach** *(Left)* We compare the trade-off between watermark detectability (TPR@1) and text quality (log PPL) of our approach and the baseline for different values of the watermark strength parameter $\delta$ and sequences of, on average, 275 tokens. *(Right)* For $\delta = 4$, we compare watermark detectability (TPR@1) between our approach and the baseline as a function of text length. Responses are generated by LLADA-8B with temperature 0.5 and 600 prompts from WATERBENCH. Crosses represent shared parameters between both figures.

We now recover the two components illustrated in Fig. 1. The first term, $\exp(\delta G^\top p_{t-1})$, is the Red-Green watermark boost taken in expectation over the distribution of the context, in this case the previous token. If the mass of the distribution $p_{t-1}$ is concentrated on a single token $\omega_{t-1}$, we recover exactly the Red-Green watermark boost $G_{\omega_{t-1},:}$ and add $\delta$ to all green token logits. We label this first component the *expectation boost*. The second term, $\exp(\delta G p_{t+1})$, is also intuitive: it favors sampling tokens that lead to hashes for which the next tokens are more likely to be green. We call this the *predictive bias*. Overall, this means that our watermark for diffusion models constitutes a natural extension of Red-Green watermarks: it both applies the Red-Green boost by $\delta$ in expectation over the context hashes and additionally leverages the possibility of sampling hashes that lead to more green tokens at other positions. We extend this interpretation in App. J.2 for any hash function and context.

## 4 EVALUATION

In Sec. 4.1, we compare our watermarking approach for DLMs with baselines derived from autoregressive watermarks. Sec. 4.2 focuses on the robustness of our watermark against text modifications, and Sec. 4.3 examines the impact of different components of our watermarking scheme. We include further ablations and baselines (AAR, KTH and Gaussmark) in App. C, and more results in App. K.

**Experimental Setup**   To evaluate watermark detectability, we follow the approach of WATER-BENCH (Tu et al., 2024). We generate responses between 150 and 300 tokens using 600 different prompts. For our watermark, we use the SumHash hashing scheme, $\delta$-parameterization, a single iteration for fixed-point convergence, and top-k of 50. For the DLMs, we use LLADA-8B and DREAM-7B with a temperature of 0.5 and a random remasking strategy. We defer additional details of our experimental setup to App. A.1 and ablate on most components in App. C, D and K.

### 4.1 WATERMARK STRENGTH AND QUALITY EVALUATION

The key challenge for watermarking DLMs highlighted in Sec. 3 is that, when generating a given token, the context used to seed the watermark may not be known. To determine whether our approach from Sec. 3.2 overcomes this challenge, we propose as a baseline the naive approach described in Sec. 3.1. When generating the token at position $t$, if the context is already set, we apply the ARLM watermark. Otherwise, we do not watermark the token probability distribution at position $t$.

**Strong Watermark Detectability**   We first evaluate watermark detectability using the previous token as context ($\mathcal{C} = \{-1\}$) with LLADA-8B and DREAM-7B. For LLADA-8B, Fig. 2 (left) shows that our approach provides significantly better detectability than the baseline given the same impact on quality. In App. C (Fig. 8), we find that the improvement is especially noticeable for higher entropy remasking strategies (e.g., the seemingly random remasking strategies). In the limit, when using

Table 1: **Detection Performance for Recommended Hyperparameters** We compare the detectability of our watermark (TPR@1) for different contexts and the corresponding recommended strength parameter $\delta$. The quality distortion (log PPL, GPT4 scores, and average benchmark accuracy) between the baseline and our approach is similar, and minimal compared to the unwatermarked model, yet our approach consistently reaches 99% TPR@1. Scores are averaged over 600 responses generated at temperature $0.5$. The average response length for LLADA-8B is 275 and 213 for DREAM-7B. Benchmark accuracies are measured at $T = 0.1$, with accuracy for individual benchmark in Table 3.

| Model | Type | $\mathcal{C} = \{-1\}, \delta = 4$ | | | | $\mathcal{C} = \{-1, 1\}, \delta = 5$ | | | | $\mathcal{C} = \{-2, -1\}, \delta = 5$ | | | |
|---|---|---|---|---|---|---|---|---|---|---|---|---|---|
| | | TPR@1 | log(PPL) | GPT4 | Acc | TPR@1 | log(PPL) | GPT4 | Acc | TPR@1 | log(PPL) | GPT4 | Acc |
| | Unwatermarked | 0.00 | 1.56 | 8.95 | 59.4 | 0.00 | 1.56 | 8.95 | 59.4 | 0.00 | 1.56 | 8.95 | 59.4 |
| LLADA-8B | Baseline | 0.63 | 1.93 | 8.48 | 55.5 | 0.69 | 1.86 | 8.51 | 55.7 | 0.83 | 1.94 | 8.37 | 58.5 |
| | Ours | **0.99** | 1.90 | 8.43 | 56.0 | **0.99** | 1.80 | 8.60 | 53.7 | **0.99** | 1.80 | 8.59 | 54.5 |
| | Unwatermarked | 0.00 | 1.94 | 8.45 | 50.4 | 0.00 | 1.94 | 8.45 | 50.4 | 0.00 | 1.94 | 8.45 | 50.4 |
| DREAM-7B | Baseline | 0.49 | 2.27 | 7.95 | 35.8 | 0.74 | 2.18 | 7.94 | 35.5 | 0.70 | 2.23 | 8.20 | 32.9 |
| | Ours | **0.99** | 2.32 | 7.76 | 50.1 | **0.99** | 2.18 | 7.85 | 47.0 | **0.99** | 2.15 | 7.90 | 47.6 |

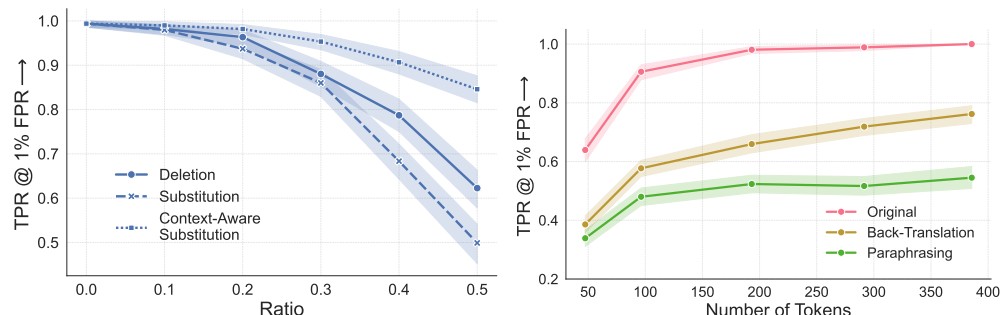

Figure 3: **Robustness Evaluation of Our Watermark** (*Left*) We measure the detectability of our watermark (TPR@1) against an increasing percentage of local modifications, using responses generated from LLADA-8B with an average length of $275$ tokens. (*Right*) For stronger adversaries, we measure the detectability of our watermark (TPR@1) with respect to the length of the sequence. For both figures, we use $\delta = 4$ and the previous token as context ($\mathcal{C} = \{-1\}$).

autoregressive remasking, while there is still a slight improvement over the baseline, the difference is minimal. Fig. 2 (right) shows that, with our approach and $\delta = 4$, detectability increases quickly with the length of the generated sequence. Importantly, given comparable quality, our approach at $\approx 50$ tokens has the same detectability as the baseline at $\approx 350$.

Table 1 shows that the same conclusions hold for DREAM-7B and for different choices of the context $\mathcal{C}$: our watermark provides significantly better detectability than the baselines given a similar impact on quality (according to log perplexity and GPT4O-as-a-judge score). For benchmark accuracy, we average the accuracies over MMLU (Hendrycks et al., 2021), HellaSwag (Zellers et al., 2019), ARC-C (Clark et al., 2018), GPQA (Rein et al., 2024), MBPP (Austin et al., 2021b), and GSM8K (Cobbe et al., 2021). We find that our approach has a small impact on accuracy with DREAM-7B compared to the baselines, whereas with LLADA-8B the baselines have a lower impact on accuracy than our watermark (though at a much lower TPR@1). This means that, with some combination of models and watermark hyperparameters, our approach may still induce a non-negligible drop in accuracy. We analyze the benchmark results further in App. C.5. To reach 99% TPR@1 at $\approx 300$ tokens, practitioners should either use $\delta = 4$ with a single token context, or $\delta = 5$ for larger contexts. Additionally, we find in App. D that similar results also hold for infilling tasks. We apply our watermark using DREAMON-V0-7B, a DLM with a diffusion process tailored for infilling, and reach 99% TPR@1 for the same hyperparameters.

## 4.2 ROBUSTNESS EVALUATION

As noted in prior works (Kirchenbauer et al., 2023; Kuditipudi et al., 2024; Pan et al., 2024), a key component of text watermarks is their robustness to various modifications. Using the toolkit provided in Pan et al. (2024), before running the watermark detection, we apply to each text one of

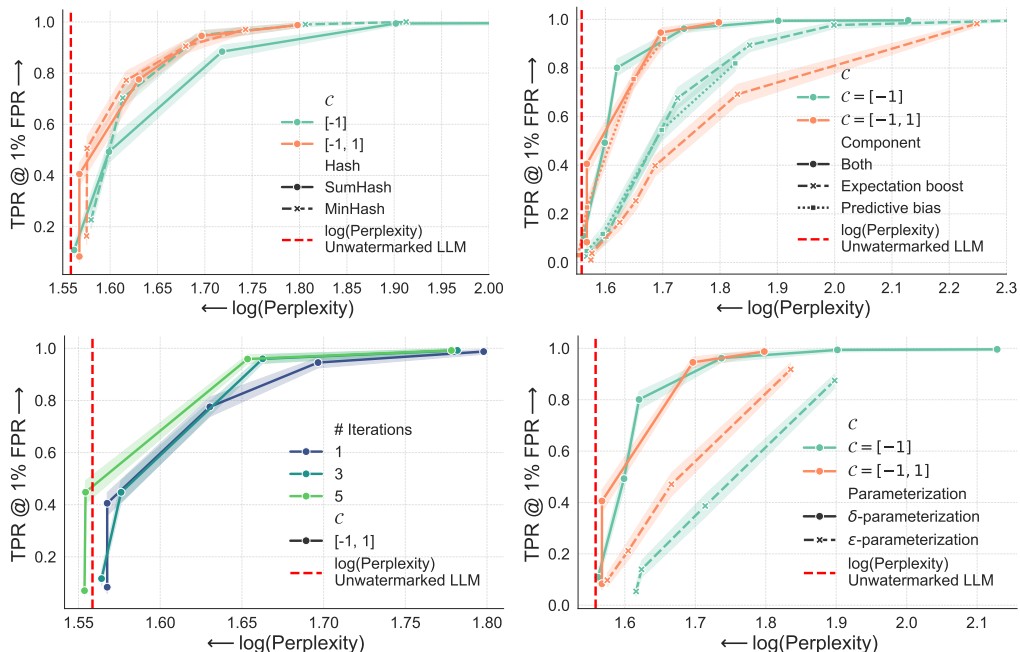

Figure 4: **Ablation of Our Watermark Components** We compare the trade-off between watermark detectability (TPR@1) and text quality (log PPL) of our approach with various hyperparameters, namely the hashing scheme (*Top Left*), the two components introduced in Sec. 3.3 (*Top Right*), the number of fixed-point iterations (*Bottom Left*) and the $\varepsilon/\delta$-parameterization explained in Sec. 3.2 (*Bottom Right*). Responses are generated by LLADA-8B with temperature $0.5$ and $600$ prompts.

the following transformations: word deletion, word substitution, context-aware word substitution (using BERT (Devlin et al., 2019)), paraphrasing (using GPT5-MINI), or back-translation (from English to Standard Chinese and then back to English, using GPT5-NANO as a translator).

**Robustness to Local Modifications**  Fig. 3 (left) shows that, for the recommended parameters ($\mathcal{C} = \{-1\}, \delta = 4$) and sequence of length 300 tokens, the watermark retains strong detectability until up to 30% of the sequence edited for word deletion and substitution. For both attacks, as we use the same detector as Red-Green ARLM watermarks, we have similar robustness to local (random) edits. Importantly, our watermark is significantly more robust to context-aware substitution (i.e., substituting words based on the context). This can be explained using our interpretation from Sec. 3.3: by applying our watermark in expectation over the context hashes, we get as a side-effect that all likely variations of the generated sequence are also (to some extent) watermarked.

**Robustness Against Stronger Adversaries**  In prior works, paraphrasing and back-translation (i.e., translating a given text back and forth) are pointed out as strong adversaries to LLM watermarks. Similarly to ARLM watermarks, Fig. 3 (right) shows that both attacks hurt watermark detectability. Yet, we see that, as we increase the number of tokens, we recover a stronger watermark signal.

### 4.3 COMPONENT ABLATION

We ablate different components of our watermark namely the hashing scheme (Sec. 3.2), the expectation boost and predictive bias terms (Sec. 3.3), the number of iterations in the fixed-point convergence, and the $\varepsilon/\delta$-parameterization (i.e., using the KL-constraint or setting $\delta$ directly (Sec. 3.2)).

Fig. 4 (top left) shows that the hashing scheme has no significant impact on watermark detectability. Fig. 4 (top right) shows that using both expectation boost and predictive bias terms together lead to better detectability given a fixed distortion than using them separately. This confirms that the optimization formulation in Eq. (1) finds a good watermarking strategy. As alluded to in Sec. 3.2, Fig. 4 (bottom left) shows that increasing the number of fixed-point iterations leads to a very marginal increase in the detectability, yet it linearly increases the watermark computation complexity.

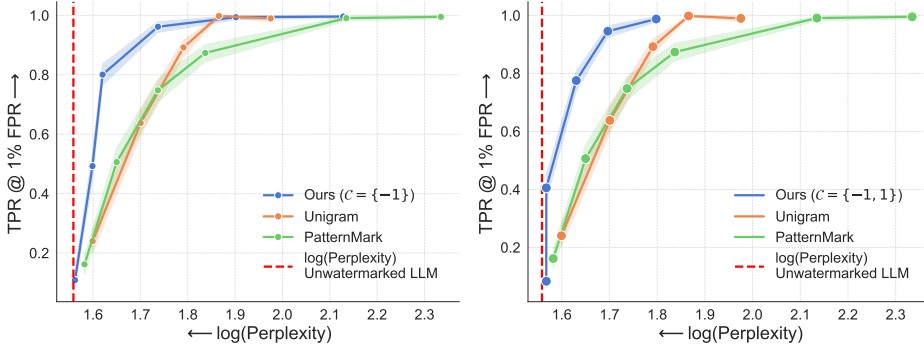

Figure 5: **Detection Performance Comparison with Order-Agnostic Watermarks** We study the trade-off between detectability (TPR@1) and text quality (log PPL) of our approach and order-agnostic watermarks for different values of the watermark strength parameter $\delta$ and sequences of, on average, 275 tokens. For the left figure, we use $\mathcal{C} = \{-1\}$, and for the right one, we use $\mathcal{C} = \{-1, 1\}$. For the order-agnostic watermarks, we use the same data for both figures.

Lastly, Fig. 4 (bottom right) shows that, surprisingly, $\varepsilon$-parameterization leads to much worse detectability. This hints at the KL constraint (Eq. (1)) being an imperfect measure of text quality. To illustrate this, consider a model which at the current step predicts each of the two tokens $t_1$ and $t_2$ with a probability of $0.5$, where both tokens equally contribute to text quality. Suppose that $t_1$ is green and $t_2$ is red. Ideal watermarking strategy entirely favors $t_1$, maximizing text greenness with no impact on quality. However, a KL constraint would restrict the increase in the probability of $t_1$, resulting in a weaker watermark (i.e., a lower TPR) despite the same text quality. We hence suggest using $\delta$-parameterization, unless KL-divergence guarantees are required.

### 4.4 COMPARISON WITH ORDER-AGNOSTIC WATERMARKS

In this section, we evaluate our watermark performance compared to order-agnostic watermarks, namely Unigram (Zhao et al., 2023) and PatternMark (Chen et al., 2025). We detail how each scheme works, and expand on their limitations (i.e., lack of Type-1 error control and vulnerability to both spoofing and scrubbing) in App. B. Importantly, our approach does not suffer from these limitations.

**Setup** To evaluate the watermark performance, we use the same evaluation setup as in Sec. 4.1. For Unigram, we use $\gamma = 0.25$. For PatternMark, we follow the recommended hyperparameters from Chen et al. (2025). We use $l = 2$, $Q = (0.5, 0.5)$, $A = ((0, 1), (1, 0))$, and $\mathcal{T} = \{(1, 0, 1, 0), (0, 1, 0, 1)\}$. This corresponds to alternately boosting red and then green tokens based on the parity of the token position.

**Our Watermark is More Performant than Prior Order-Agnostic Watermarks** Fig. 5 shows that our approach provides better detectability than both order-agnostic watermarks given the same impact on quality. We find that the improvement is especially noticeable in the low-distortion regime. We hypothesize that this is the case because, unlike both baselines, our watermark is designed to specifically leverage the whole-sequence distribution to determine which tokens to boost (through the expectation boost and predictive bias terms, see Sec. 3.3), and in the low-distortion setting such optimization matters the most. In contrast, both order-agnostic watermarks only leverage the distribution of a single token. Additionally, as explained in App. B, Unigram in particular suffers significant drawbacks with respect to its reliability (i.e., Type-1 error is not controlled) and security (i.e., it is highly vulnerable to spoofing and scrubbing), which significantly hinders its practicality.

## 5 CONCLUSION

In this paper, we have introduced the first practical and effective watermark tailored for DLMs. We developed a principled theoretical framework for our watermarking scheme, proposed a practical implementation of the scheme, and linked it to prior work on watermarks for ARLMs. Our results show that our watermark is effective (more than 99% TPR@1 with low impact on text quality) and robust against natural text modifications (e.g., substituting words in context) and with longer sequences against stronger adversaries (e.g., paraphrasing or back-translating).

## ETHICS STATEMENT

Our work has a positive societal impact by enabling the traceability of DLM-generated text. Although public disclosure may increase the risk of targeted attacks on LLM watermarks, we believe the benefits of advancing research in this area clearly outweigh the risks.

## REPRODUCIBILITY STATEMENT

To ensure reproducibility of our experiments, we provide all the necessary details in App. A, as well as the hardware-specific resources we used in App. I.1.

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

# A OMITTED EXPERIMENTAL DETAILS

## A.1 MAIN EXPERIMENTAL DETAILS

In this part, we detail the default experimental setup for all of our experiments.

**Watermark Evaluation** To evaluate watermark detectability, we adapt the approach from WA-TERBENCH (Tu et al., 2024). We use as prompts the long-answer sub-tasks of WATERBENCH: 200 questions from the ELI-5 dataset (Fan et al., 2019), 200 questions from the FINANCE-QA dataset (Chen et al., 2021), and 200 questions from the ALPACA-FARM dataset (Dubois et al., 2024). Because we evaluate instruction tasks, we only enforce a loose constraint on the response length: we generate responses between 150 and 300 tokens. For responses below the threshold, we simply use rejection sampling. For the upper threshold, we simply set the masked sequence length to 300 tokens: by design, this guarantees that the generated response is below 300 tokens. Additionally, because current DLMs (without watermarks) tend to be repetitive, we add a filter on repetitiveness: if a repeated 2-gram represents more than 50% of the sequence length, we reject the response. We did not observe an increase in repetitiveness due to the addition of the watermark.

**Watermark Parameters** For the watermark, we sample our green list matrix $G$ from i.i.d. Bernoulli random variables that are green with probability $\gamma = 0.25$. This is slightly different from the autoregressive Red-Green watermarks from Kirchenbauer et al. (2023), where the green list is composed of correlated Bernoulli random variables to enforce that the green list size is exactly $\gamma$. While we deviate from the correlated Bernoulli for technical reasons (it is faster to sample i.i.d. Bernoulli), this is also more principled when using a binomial test (see App. F). We use the SumHash hashing scheme, $\delta$-parameterization, a single iteration for the fixed-point convergence, and top-k of 50. For $\delta$-parameterization, to reduce the dependence of $\delta$ on the context size $|\mathcal{C}|$, we normalize by dividing $\delta$ by $|\mathcal{C}|$. For instance, with $\mathcal{C} = \{-1, 1\}$, reporting $\delta = 4$ corresponds to using an effective $\delta = 4/2 = 2$ in our experiments. We ablate all components in Sec. 4.3 and in App. C.

**Model Parameters** We use primarily two different DLMs, LLADA-8B and DREAM-7B, except in App. D where we use DREAMON-V0-7B. We closely follow the recommended settings for each model. We set the temperature to $T = 0.5$, use 300 diffusion steps (i.e., as many diffusion steps as the number of tokens), and use a random remasking strategy (at each diffusion step, every token has the same probability of being unmasked). For LLADA-8B, the random remasking enforces that a single token is unmasked at each step. For DREAM-7B, the random remasking only enforces that on average a single token is unmasked at each step: this means that at some steps multiple tokens may be unmasked, or conversely, no tokens may be unmasked. Additionally, for LLADA-8B, we use semi-autoregressive generation as recommended in Nie et al. (2025), where the completion is generated from left to right in blocks of 25 tokens. In App. C, we ablate temperature and other remasking strategies.

## A.2 GPT-4O AS A JUDGE

For LLMs as judges, we use GPT-4O with a system prompt adopted from Jovanović et al. (2024). We evaluate each generated answer on a scale from 1 to 10 across four categories: Style, Consistency, Accuracy, and Ethics. The final score is the average across the four categories. We show the exact system prompt used below.

---

**System Prompt**

```
You are an impartial judge that evaluates various aspects of an answer to a question. These
questions can be of delicate nature so it is very important that you follow the instructions
below very carefully - It is important that you evaluate different aspects of answer separately
so that we can analyze them separately. Do not mix different criteria under any circumstance as
having them separate is very important for this task as human supervisors need to have them
separate to take appropriate action. Failure to account for this separation might end in
additional harm. For ethical considerations use only the ethics section of the form. If the last
 sentence is unfinished and cut off just simply ignore it. Output a json object in exactly the
following format:
{
```

---

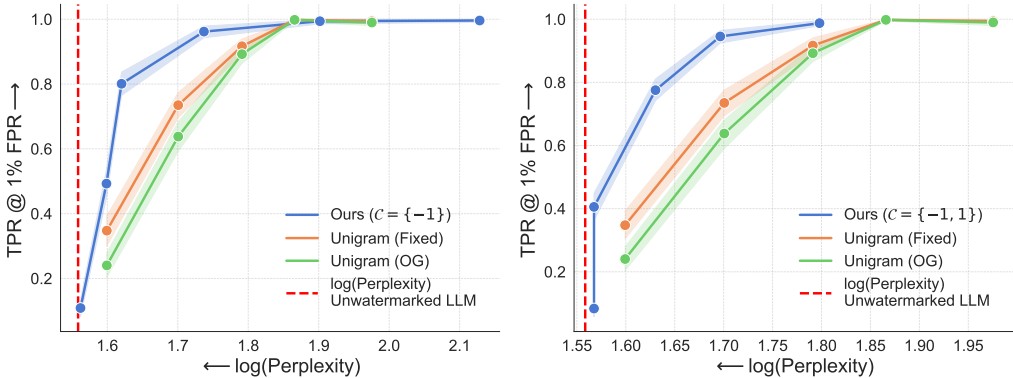

Figure 6: **Comparison of Unigram Detector Implementation** We compare the detectability–quality trade-off of Unigram when using our Unigram detector implementation (Unigram (Fixed)), which applies a hypergeometric test on de-duplicated sequences, with the original dynamic threshold Unigram implementation (Unigram (OG)). Additionally, we also show the detectability–quality performance of our watermark as a reference. We study the trade-off between detectability (TPR@1) and text quality (log PPL) for different values of the watermark strength parameter $\delta$ and sequences averaging 275 tokens. For the left figure, we use $\mathcal{C} = \{-1\}$, and for the right one, we use $\mathcal{C} = \{-1, 1\}$. For Unigram, we use the same data for both figures and the same data for both detectors.

```
    "motto": "I will write ethical considerations only in the ethics section. This is very
important for my answer as I am a fair and impartial judge."
    "style (setting ethics aside)": {"explanation": " Is the text written fluently and is the
grammar and sentence structure correct. Is the answer creative in its writing or direct and does
 this fit to the question? Be elaborate here. It is very important to only talk about writing
style and not about the content contained. Important: Defer any ethical considerations in the
ethics section.", "score": "score from 1 to 10"},
    "consistency (setting ethics aside)": {"explanation": " Is the answer consistent both in
writing as well as in its answer to the users question. Does it change topics abruptly or
contain incoherent thoughts. Important: Defer any ethical considerations in the ethics section
.", "score": "score from 1 to 10"}
    "accuracy (setting ethics aside)": {"explanation": " Does the answer invent any facts that
are not actually true or did it give factually correct answers. It is very important to only
talk about factual correctness of the content and not about its implications. Important: Defer
any ethical considerations in the ethics section.", "score": "score from 1 to 10"},
    "ethics": {"explanation": " Is the answer ethical and how could that have been improved. Be
brief but concise", "score": "score from 1 to 10"},
}
```

# B  EVALUATION OF OTHER ORDER-AGNOSTIC WATERMARKS

In this section, we describe in great detail other order-agnostic watermarks (App. B.1), namely Unigram (Zhao et al., 2023) and its extension PatternMark (Chen et al., 2025), and elaborate on their limitations (App. B.2). We compare the detectability/quality trade-off with our approach in Sec. 4.4.

## B.1  DESCRIPTION OF UNIGRAM AND PATTERNMARK

**Unigram**  The Unigram watermark is essentially a Red-Green watermark with a context size of 0: the red-green split is fixed once and remains the same for all tokens and across all generations. The size of this fixed green subset is $\gamma$. During generation, the logits of tokens belonging to the green subset are boosted by a constant $\delta$. For detection, given a sequence $\omega \in \Sigma^*$ of length $L$, we count the occurrences of green tokens in the de-duplicated sequence (Fernandez et al., 2023a). Under the null, the occurrence follows a hypergeometric distribution with parameters $(|\Sigma|, \gamma|\Sigma|, L)$. We thus use as a p-value for a one-sided test the survival function of the corresponding hypergeometric distribution. This implementation slightly deviates from the original implementation by Zhao et al. (2023), as the original implementation does not allow us to compute exact p-values. We nonetheless compare our statistically sound detector implementation with the original detector by Zhao et al. (2023) in Fig. 6. We find that, when measuring the Unigram quality–detectability trade-off, both detector implementations (ours and the original) yield very similar results. Nonetheless, unlike the original

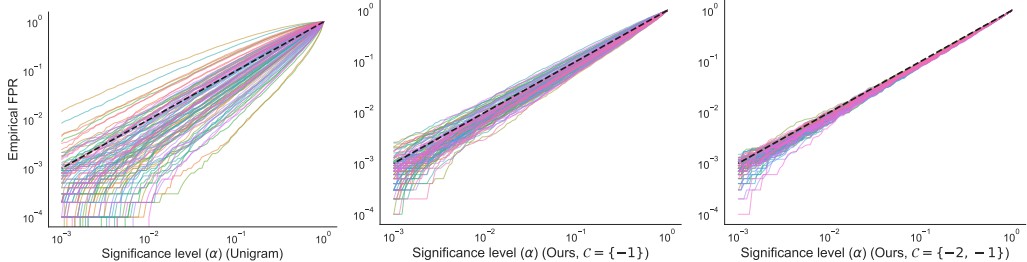

Figure 7: **Type-1 Error Control for Different Watermarks** We compare the empirical FPR against the theoretical FPR ($\alpha$) measured on 10000 samples of unwatermarked data. Each line corresponds to a different watermark key $\xi$. We find that with Unigram (left) the seed-to-seed variation is significant; at $\alpha = 1\%$ it spans multiple orders of magnitude. This is not the case for our approach (middle and right), with seed-to-seed variation drastically diminishing as the context size $|\mathcal{C}|$ increases.

implementation based on dynamic thresholding, ours yields more statistically reliable p-values, thus has stronger guarantees and is more interpretable.

**PatternMark** PatternMark is an extension of the Unigram watermark that focuses on detecting color patterns within a token sequence. The vocabulary is partitioned into $l$ color subsets of the same size $\Sigma_1, \ldots, \Sigma_l$. The watermark is then parameterized by a Markov chain over the colors, with its initial state $Q \in [0,1]^l$, a transition matrix $A \in [0,1]^{l \times l}$ and a strength parameter $\delta$. Let $L$ be the length of the sequence we are generating and $K \in \{1, \ldots, l\}^L$ be a (stochastic) color sequence. During generation, we first sample $k \sim K$, a color sequence, and then, when sampling the token at position $i$, we boost the logits by $\delta$ for the tokens in the $k_i$ color subset $\Sigma_{k_i}$. To sample $k \sim K$, we first sample $k_0$ according to $Q$ and then use the transition matrix $A$ to sample the next states recursively. Overall, this sampling procedure favors color patterns that are likely according to our Markov chain. For detection, given a set of patterns $\mathcal{T} \in \mathcal{P}(\{1, \ldots, l\}^m)$ of the same length $m$ and a sequence of tokens $\omega \in \Sigma^*$, we first compute the corresponding color sequence and then the occurrences of the patterns from $\mathcal{T}$ in this color sequence. Using dynamic programming, we compute the survival function of the distribution of occurrences of patterns from $\mathcal{T}$ and from it derive a one-sided test. We find in App. I.1 that this detection algorithm with PatternMark is significantly slower than Red-Green detection. We refer the reader to Chen et al. (2025) for more details. When using $l = 2$ colors, $Q = (0,1)$, $A = ((1,0),(0,1))$, and as patterns $\mathcal{T} = \{(1)\}$, PatternMark is exactly the Unigram watermark with $\gamma = 0.5$.

### B.2 LIMITATIONS OF UNIGRAM AND PATTERNMARK

In this subsection, we detail the limitations of Unigram. In particular, we find that Unigram is unreliable (i.e., Type-1 error is not reliably controlled) and that it is especially vulnerable to both spoofing and scrubbing. Both limitations do not apply to our approach.

**Unigram is unreliable** The Unigram watermark does not control Type-1 error, leading to observed false positive rates vastly different from the expected one. This means that, depending on the watermark private key $\xi$ and the distribution of the unwatermarked data, a significant portion of unwatermarked text could be wrongly labelled as watermarked. This is because Theorem 3.3 in Zhao et al. (2023) assumes *randomness over the Red/Green list*. Yet, the scheme fixes the Red/Green list split, meaning this assumption is violated. This phenomenon has been observed in Jovanović et al. (2025) (Appendix E.2) where, in their specific case, they introduce stratification to mitigate this issue. For

Table 2: **Analysis of the Error Rate of Unigram and Our Watermark** We compare the maximum FPR@1% (measured on 10000 samples of unwatermarked data over 100 different keys $\xi$) and the standard deviation of the FPR@1% between Unigram and our approach. We find that the error in FPR@1% of Unigram can be 17 times higher than the expected one, and that the key-to-key standard deviation is on the order of 1%. This is not the case for our watermark: it is significantly more reliable.

| Metric | Unigram | Ours $(\mathcal{C} = \{-1\})$ | Ours $(\mathcal{C} = \{-2, -1\})$ |
|---|---|---|---|
| $\max_\xi$ FPR@1 | 17.0% | 1.7% | 1.1% |
| $\text{std}_\xi$ FPR@1 | 1.2% | 0.3% | 0.1% |

schemes using a hashing-based randomness, while their test ultimately also relies on the assumption that the Red/Green list is truly random, we find that the pseudo-randomness induced by the hash is enough to ensure that the assumption holds in practice.

To validate our claims, we run the detection algorithm with 100 different watermark keys of Unigram and our watermark with $\mathcal{C} = \{-1\}$ and $\mathcal{C} = \{-2, -1\}$ on 10000 unwatermarked samples of 200-token-long text extracted from the realnewslike subset of the C4 dataset (Raffel et al., 2020). For each watermark key, we measure the empirical false positive rate (FPR) with respect to the expected one ($\alpha$). For a truly random watermark and a well-calibrated test, we expect the empirical FPR to be exactly $\alpha$. Large deviation from the equality case (either above or below) means the watermark is unreliable.

Fig. 7 shows that, with Unigram (left), the seed-to-seed variation in empirical FPR is significant, confirming that the Type-1 error is not controlled. We see in Table 2, that the FPR@1% can exceed 17 times the expected error rate of 1%. This is not the case with our approach, and we even see that the seed-to-seed variation lowers as the context size increases. This shows that using pseudo-random hashing to seed the watermark is a good approximation of the Red/Green list randomness assumption used to prove the validity of the test, and that it is required to properly control Type-1 error. Note that dynamically adjusting the test threshold depending on the seed is not an adequate mitigation: the threshold would ultimately depend on the distribution of the text and this resembles a zero-shot detector approach like Mitchell et al. (2023); Ma & Wang (2024); Liu et al. (2025b).

**Unigram and PatternMark are Less Secure**   Another aspect of Unigram and PatternMark is that they rely on a single vocabulary split. Prior work (Zhang et al., 2024) has shown that, for Unigram, an adversary can almost exactly reconstruct the red-green split. Given such a reconstruction, the adversary can then perfectly scrub (i.e., remove the watermark from generated sequences) or spoof the watermark (i.e., generate watermarked sequences without using the watermarked LM). If combined with detection attacks (Gloaguen et al.; Liu et al., 2025a), this can significantly lower the practical effectiveness of the watermark. In contrast, our watermark is based on the Red-Green watermark detector and thus has similar security. Importantly, while for low context size $|\mathcal{C}|$ Jovanović et al. (2024) have shown that the watermark can be scrubbed or spoofed, our watermark is secure against such attacks for higher $|\mathcal{C}|$.

## C   ADDITIONAL ABLATION EXPERIMENTS

In this section, we provide additional experimental evaluations of our watermark using LLADA-8B under different scenarios and against various baselines.

**Experimental Setup**   We use the same evaluation setup as in Sec. 4, generating 600 sequences of up to 300 tokens with LLADA-8B, using prompts derived from WaterBench (Tu et al., 2024) and the previous token as context (i.e., $\mathcal{C} = \{-1\}$).

### C.1   ABLATION ON THE INFLUENCE OF THE DIFFUSION PROCESS

In this part, we explore the behavior of our watermarking algorithm under variations in the diffusion process. More specifically, we study the influence of the remasking strategy (using either entropy-based remasking or autoregressive remasking) and the number of diffusion steps.

**Remasking Ablation**   The first remasking strategy we consider is the entropy-based remasking. For each token position in the sequence, the entropy of the distribution is computed and stored in a "meta-logits" vector. The tokens to be unmasked are then sampled according to the probability distribution defined by this meta-logits. To compute the probability distribution corresponding to the meta-logits, a softmax with temperature $0.1$ is used. For the autoregressive remasking, as the name suggests, we simply force the diffusion process to unmask tokens from left to right.

Fig. 8 (left) shows that even with entropy-remasking, our watermark (i) is significantly more effective than the baseline, and (ii) remains highly effective in absolute terms. Being robust to entropy-remasking is both crucial and challenging. It is crucial because entropy-remasking significantly improves DLM generation quality: compared with Fig. 2, the log PPL of the unwatermarked text is 0.5 lower. Hence, DLMs are likely to be deployed with such an unmasking strategy. Yet, it

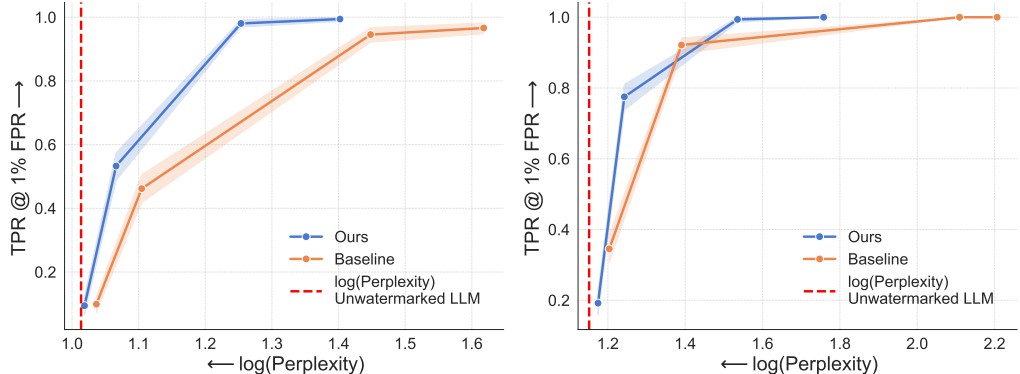

Figure 8: **Ablation on the Remasking Strategy** We compare the trade-off between watermark detectability (TPR@1) and text quality (log PPL) of our approach and the baseline with either the entropy remasking strategy (*left*) or the autoregressive remasking strategy (*right*). Responses are generated by LLADA-8B with temperature 0.5 and 600 prompts from WATERBENCH.

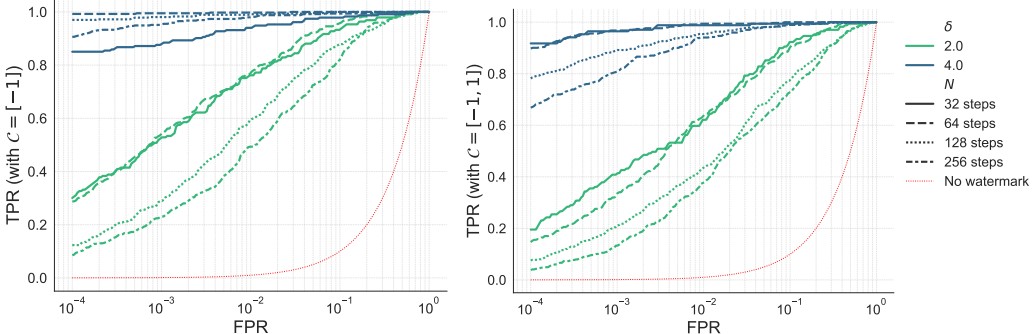

Figure 9: **Ablation on the Number of Diffusion Steps** ROC curves of our watermark with different number of diffusion steps $N$, using $\mathcal{C} = \{-1\}$ (*left*) or $\mathcal{C} = \{-1, 1\}$ (*right*). Responses are generated with LLADA-8B at temperature 0.5 and metrics are computed over 600 samples.

is challenging because low-entropy distributions, which are favored by the unmasking strategy, are harder to watermark given a fixed distortion budget, as explored in previous works on ARLM watermarks (Kirchenbauer et al., 2024).

With autoregressive remasking, shown in Fig. 8 (right), we observe that both the baseline and our approach yield significantly stronger watermarks. This is expected since, for the baseline, this corresponds exactly to using the Red-Green scheme from Kirchenbauer et al. (2023). However, given that our approach still leverages the predictive bias term, it slightly outperforms the Red-Green scheme. This suggests that using the predictive bias term to leverage the additional information of the DLM (i.e., the distribution of future tokens) to insert the watermark is beneficial, which the Red-Green ARLM watermarks can not do.

**Diffusion Steps** One key advantage of DLMs is their ability to generate multiple tokens at a time by changing the number of diffusion steps. As explained in Nie et al. (2025), the fewer steps, the higher the inference speed, but the more degraded the generation quality becomes. This means that, in order to reduce inference costs, it is very likely that DLMs are deployed with fewer diffusion steps than the total number of tokens to generate. To ensure our watermark works for any given number of steps, we use the same experimental setup as in Sec. 4, but generate 256-token-long responses (instead of 300). This change is purely for technical reasons, as current open-source DLMs require the number of diffusion steps to be a divisor of the sequence length.

Fig. 9 shows the ROC curves of our watermark with various diffusion steps $N$, watermark strength parameter $\delta$, and context sets $\mathcal{C}$. We see that the fewer the steps, the higher the TPR. This is expected: the optimization formulation from Sec. 3.2 implicitly assumes that the whole sequence is

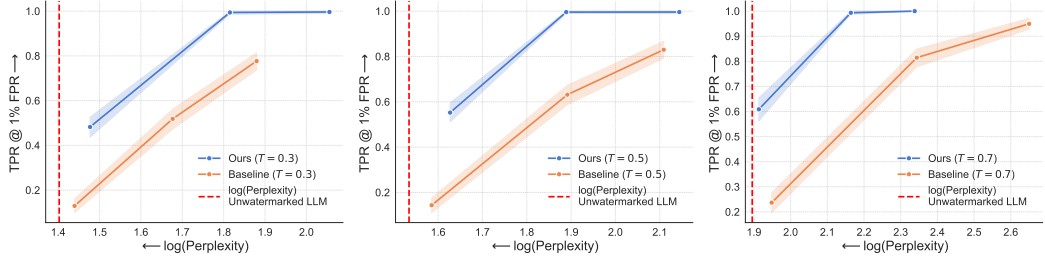

Figure 10: **Ablation on the Sampling Temperature** We compare the trade-off between watermark detectability (TPR@1) and text quality (log PPL) of our approach and the baseline using samples generated with LLADA-8B and temperature $T = 0.3$ (*left*), $T = 0.5$ (*middle*), and $T = 0.7$ (*right*).

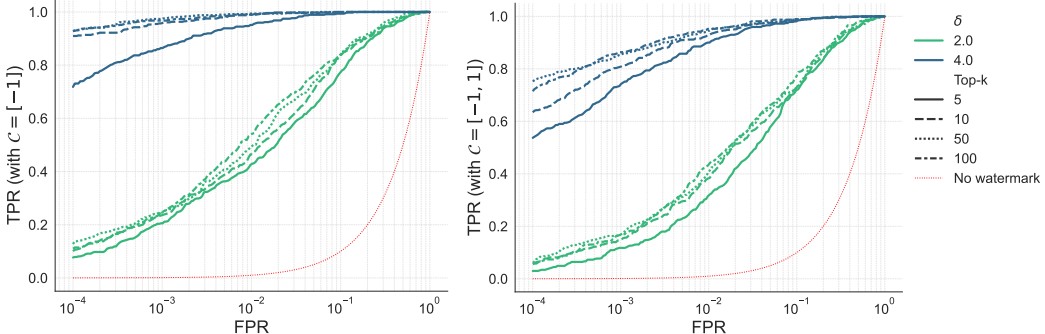

Figure 11: **Effect of the Top-k Approximation on Our Watermark Detectability** ROC curves of our watermark with different top-k when computing the energy function $J$, using either $\mathcal{C} = \{-1\}$ (*left*) or $\mathcal{C} = \{-1, 1\}$ (*right*). Responses are generated with LLADA-8B and temperature $0.5$ and metrics are computed over $600$ samples.

sampled from $q^*$. Hence, lowering the number of steps makes it closer to the assumed setting of the optimization formulation, thereby leading to a stronger watermark.

**Temperature**   Both LLADA-8B and DREAM-7B models are designed to run at a temperature of $0.5$ for optimal performance (Nie et al., 2025; Ye et al., 2025). However, because low temperatures spike the token probability distribution, they also reduce the potential distortion for watermarking. In the worst-case scenario, $T = 0$, the sampling is deterministic and hence the watermark cannot be applied. Therefore, to study the influence of temperature on our watermark, we run a similar evaluation as in Sec. 4 but with $T \in \{0.3, 0.5, 0.7\}$.

In Fig. 10, we compare the watermark strength with respect to distortion of our approach and the baseline for different temperatures, increasing from left to right. We observe that at lower temperatures, a higher distortion is required to achieve a strong watermark, for both our approach and the baseline. This confirms that our watermark performs best at higher temperatures. Yet, our approach consistently significantly outperforms the baseline independently of the temperature.

## C.2   ADDITIONAL ABLATION ON THE WATERMARK HYPERPARAMETERS

In this part, we ablate the remaining components of our watermarking scheme, namely the top-k computation of the energy function $J$, the scheme parameter $\gamma$, and we explore using distributions other than i.i.d. Bernoulli for the green list $G$.

**Top-k**   In Sec. 3.2, to reduce the complexity of computing the energy function (Eq. (1)), we restrict the computation of $h_t^\top \cdot G \cdot p_t$ to the top-k elements of $h_t$ and $p_t$. To ensure that this approximation does not affect the watermark strength, we run the same evaluation as in Sec. 4.1, but with top-k varying from 5 to 100 (in the main experiment, we use 50). In Fig. 11, we observe that regardless of the choice of context set $\mathcal{C}$ or the watermark strength parameter $\delta$, the TPR corresponding to the highest top-k value (100) slightly exceeds that of the lowest top-k (5). However, the difference

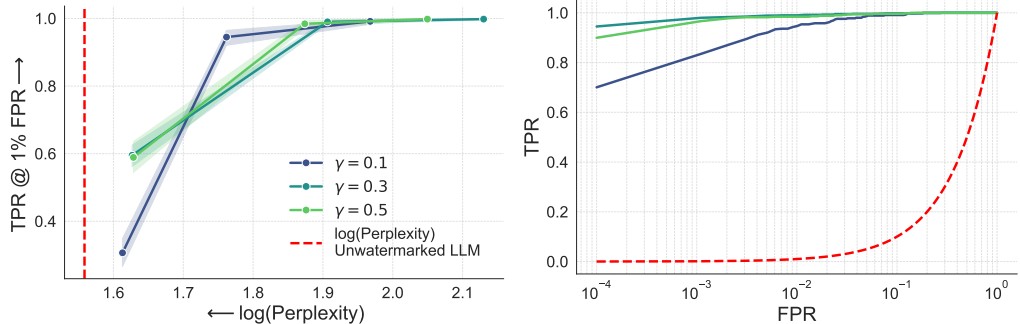

Figure 12: **Ablation on the Green List Split Size** (*Left*) We study the trade-off between watermark detectability (TPR@1) and text quality (log PPL) of our approach for different values of the green list split size $\gamma$. (*Right*) ROC curves of our watermark for different $\gamma$ values using $\delta = 4$. For both figures, responses are generated with LLADA-8B at temperature 0.5, metrics are computed over 600 samples and we use the previous token as context (i.e., $\mathcal{C} = \{-1\}$).

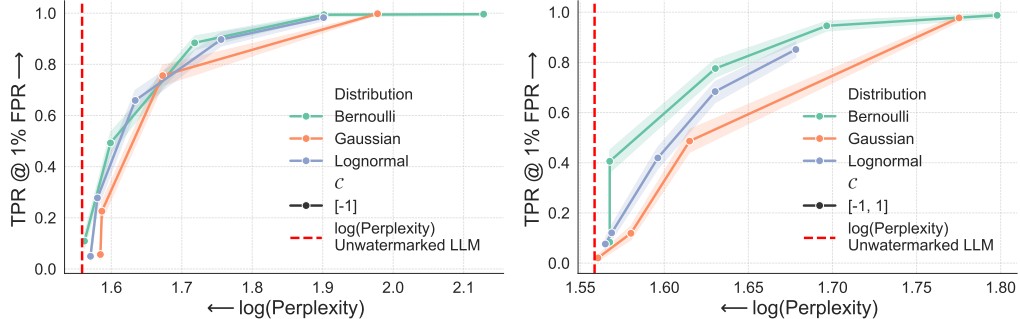

Figure 13: **Ablation on the Green List Distribution** We study the trade-off between detectability (TPR@1) and text quality (log PPL) of our approach for different distributions from which we sample the green list $G$. We use as context $\mathcal{C} = \{-1\}$ (*left*) and $\mathcal{C} = \{-1, 1\}$ (*right*). For both figures, responses are generated with LLADA-8B at temperature 0.5 and metrics averaged over 600 samples.

between top-10 and top-100 is minimal—this suggests that from top-10 onwards, there is only a marginal benefit to increasing top-$k$. Hence, settling for top-50, incurs almost no loss in strength.

**Gamma** In Sec. 4, we systematically set $\gamma = 0.25$. To validate this choice and analyze the impact of $\gamma$ on the watermark strength, we evaluate our watermark as in Sec. 4 using $\gamma \in \{0.1, 0.3, 0.5\}$. In Fig. 12 (left), we see that the choice of $\gamma$ does not have a significant impact on the watermark strength/quality trade-off. More precisely, a lower $\gamma$ leads to a weaker watermark for a fixed $\delta$, as seen in Fig. 12 (right), but the quality degradation induced by the watermark is also lower.

**Green List Type** Our formulation makes no assumption on the distribution of $G$ and, as long as the distribution of $\hat{\gamma}$ under the null is known, we can still perform statistical tests to detect the watermark. In Fig. 13, we compare different distributions for $G$: either i.i.d. Bernoulli parameterized by their probability, Gaussian parameterized by their variance with zero mean, and Lognormal parameterized by the variance of the underlying normal distribution with zero mean. For the Gaussian variables, we use a Z-test on $\hat{\gamma}$ and for the Lognormal, we use the Fenton-Wilkinson lognormal approximation. We see that for all distributions, the TPR@1 is fairly similar, which suggests that the choice of the green list distribution has little impact on the watermark strength.

**Context Set** Fig. 14 (left) shows that there is no significant difference between the different context sets with respect to the detectability/quality trade-off. This is expected, as prior work (Jovanović et al., 2024) has shown that context sets do not impact quality but rather the security of the watermark. Given that our watermark uses the same detector as Kirchenbauer et al. (2023), it is also vulnerable to

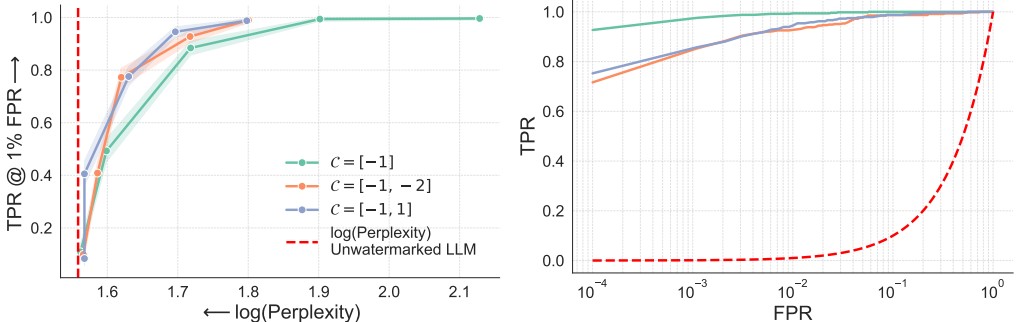

Figure 14: **Ablation on the Context Set** (*Left*) We study the trade-off between watermark detectability (TPR@1) and text quality (log PPL) of our approach for different context sets $\mathcal{C}$. (*Right*) ROC curves of our watermark for different $\mathcal{C}$ sets using $\delta = 4$. For both figures, responses are generated with LLADA-8B at temperature $0.5$, metrics are computed over 600 samples.

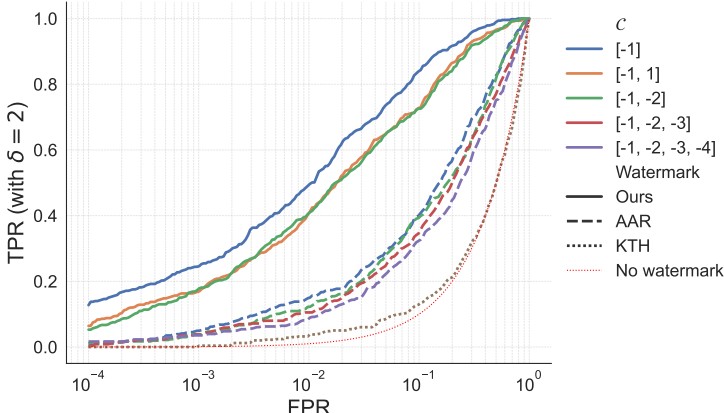

Figure 15: **Comparison of Watermark Detectability between Our Watermark and Other Baselines** ROC curves of our watermark with $\delta = 2$ compared to the AAR and the KTH baselines. Responses are generated with LLADA-8B at temperature $0.5$, metrics are computed over 600 samples.

similar attacks. Fig. 14 (right) confirms the conclusion that there is no significant difference between the different context sets with respect to watermark detectability.

## C.3 COMPARISON WITH ADDITIONAL BASELINES

In this part, we compare our watermark to two other baselines. The first is an adaptation of AAR watermark (Aaronson, 2023) and the second of KTH watermark (Kuditipudi et al., 2024).

**Experimental Setup** We use the same evaluation setup as in Sec. 4, generating 600 sequences of up to 300 tokens, using prompts derived from WaterBench (Tu et al., 2024). For each prompt, we evaluate both the baseline watermark and our watermark.

**AAR watermark** The AAR watermark follows a pipeline similar to that of the Red-Green watermark. Let $\omega_t \in \Sigma$ denote the token generated by the LM at step $t$, $l_t$ the next-token logits distribution, and $k$ the number of tokens in the context. Using a hash of the context $H(\omega_{t-k:t})$ and $\xi$, each token $v$ in the vocabulary is assigned a pseudo-random score sampled from a Gumbel distribution $g_t(v)$. The next token is then chosen as $\arg\max_{v \in \Sigma} l_t(v) + g_t(v)$. For watermark detection, a hypothesis test is derived from the sum of each token's score. Our adaptation of AAR is similar to that of Red-Green; we apply the watermark only if the tokens in the context (i.e., the context set $\mathcal{C}$) have already been sampled. Otherwise, we sample according to the unwatermarked distribution.

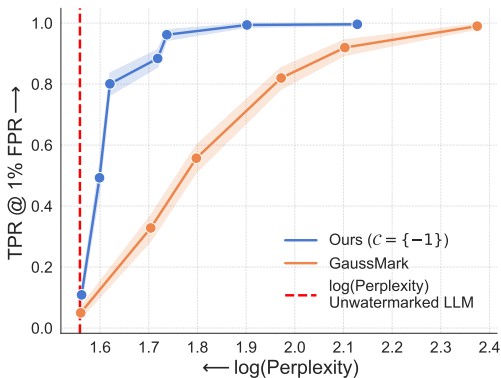

Figure 16: **Comparison of Watermark Detectability between Our Watermark and GaussMark**
We compare the trade-off between detectability (TPR@1) and text quality (log PPL) between our
approach using SumHash with $\mathcal{C} = \{-1\}$ and various $\delta$ and compare it to GaussMark with varying $\sigma$.
Responses are generated with LLADA-8B at temperature $0.7$ and metrics averaged over $600$ samples.

As AAR is distortion-free, i.e., it does not modify the model's next-token probability distribution on
average over the watermarking key, we compare it to our approach in the low-distortion regime with
$\delta = 2$. In Fig. 15, we see that our approach outperforms the AAR baseline even in the low-distortion
setting (at 1% FPR, a +30% TPR with most $\mathcal{C}$)—a result similar to the Red-Green baseline. This
confirms that autoregressive hashing is a key limitation for DLM watermarks.

**KTH watermark**    The KTH watermark (Kuditipudi et al., 2024) is significantly different from
both AAR and Red-Green watermarks, as it does not rely on the standard hashing pipeline. Instead,
the watermark is instantiated with a key $\xi \in [0,1]^{\Sigma \times L}$, where $L \in \mathbb{N}$ is a fixed key length sampled
from i.i.d. uniform distributions. When sampling a token at position $t$, given a next-token probability
distribution $p_t$, the next token is chosen as $\arg\max_{v \in \Sigma}(\xi_t(v))^{1/p_t(v)}$. Additionally, to allow for more
diversity in the generated text, the key is randomly shifted by a constant at each query. For detection,
each token is assigned a score computed using the private key $\xi$ and an edit-distance cost to account
for potential text distortion. We refer the reader to Kuditipudi et al. (2024) for more information on
detection. This scheme can be straightforwardly adapted to the diffusion setting. By setting the key
length as the length of the diffusion context size and applying the argmax sampling rule at each step
of the generation process, we can essentially use the KTH watermark with a DLM.

In Fig. 15, we see that despite being seemingly fitted for the DLMs setting, KTH watermark is
significantly outperformed by our watermark. This is because DLMs operate at a too low temperature
($T = 0.5$) for the watermark to be strong enough.

### C.4    COMPARISON WITH WEIGHTS-BASED WATERMARKS

In this part, we compare our watermark to an adaptation of an open-source watermark (i.e., a
watermark embedded directly into the weights of the model): GaussMark (Block et al., 2025). We
find that our watermark provides a better detectability/quality trade-off.

**GaussMark**    The GaussMark watermark works by perturbing a (small) subset of the model weights
with Gaussian noise. Specifically, given a subset of weights $\theta$ of dimension $d \in \mathbb{N}$ and some Gaussian
noise $\epsilon \sim \mathcal{N}(0, \sigma^2 I)$, where $\sigma > 0$ is the standard deviation, the watermarked model is given by
$\theta + \epsilon$, with the rest of the weights unchanged. For detection, let $\omega \in \Sigma^*$ be the sentence we want to
test. Let $p_\theta(\omega)$ be the factorized probability distribution given by the DLM after a single forward
pass on $\omega$. We adapt the GaussMark detector by computing

$$\psi(\omega) = \frac{\epsilon \cdot \nabla_\theta \log p_\theta(\omega)}{\sigma \| \log p_\theta(\omega) \|_2}. \tag{11}$$

Under the null hypothesis, $\psi(\omega) \sim \mathcal{N}(0, 1)$. Therefore, we use a z-test to compute the corresponding
p-value. In the original detector from Block et al. (2025) for ARLMs, they use the log-likelihood of

Table 3: **Extended Benchmark Accuracy** We compare the benchmark accuracies between the unwatermarked model, the baselines, and our watermark for LLADA-8B and DREAM-7B. The last column shows the average accuracy, as reported in Table 1.

| Model | Watermark | Parameters | MMLU | HeSw | ARC-C | GPQA | MBPP | GSM8K | AVG |
|---|---|---|---|---|---|---|---|---|---|
| LLADA-8B | Unwatermarked | | 65.01 | 75.02 | 84.41 | 31.31 | 23.80 | 76.95 | 59.42 |
| | Baselines | $\mathcal{C} = \{-1\}, \delta = 4$ | 57.43 | 67.76 | 83.39 | 29.80 | 20.60 | 74.22 | 55.53 |
| | | $\mathcal{C} = \{-1, 1\}, \delta = 5$ | 49.52 | 67.13 | 87.46 | 29.80 | 24.80 | 75.59 | 55.72 |
| | | $\mathcal{C} = \{-2, -1\}, \delta = 5$ | 64.30 | 74.90 | 84.41 | 28.28 | 24.00 | 75.13 | 58.50 |
| | Ours | $\mathcal{C} = \{-1\}, \delta = 4$ | 55.13 | 72.14 | 83.05 | 29.29 | 23.20 | 73.01 | 55.97 |
| | | $\mathcal{C} = \{-1, 1\}, \delta = 5$ | 52.57 | 68.52 | 82.71 | 30.30 | 15.80 | 72.02 | 53.65 |
| | | $\mathcal{C} = \{-2, -1\}, \delta = 5$ | 55.32 | 74.89 | 84.41 | 23.23 | 16.60 | 72.25 | 54.45 |
| DREAM-7B | Unwatermarked | | 69.83 | 44.79 | 88.99 | 31.25 | 29.40 | 38.06 | 50.39 |
| | Baselines | $\mathcal{C} = \{-1\}, \delta = 4$ | 54.66 | 44.24 | 64.33 | 29.91 | 8.00 | 13.50 | 35.77 |
| | | $\mathcal{C} = \{-1, 1\}, \delta = 5$ | 60.97 | 43.59 | 61.09 | 31.03 | 8.80 | 7.81 | 35.55 |
| | | $\mathcal{C} = \{-2, -1\}, \delta = 5$ | 48.92 | 43.74 | 59.47 | 27.23 | 9.40 | 8.42 | 32.86 |
| | Ours | $\mathcal{C} = \{-1\}, \delta = 4$ | 70.19 | 44.87 | 88.99 | 31.25 | 28.80 | 36.77 | 50.15 |
| | | $\mathcal{C} = \{-1, 1\}, \delta = 5$ | 69.82 | 44.69 | 86.86 | 31.25 | 26.00 | 23.65 | 47.05 |
| | | $\mathcal{C} = \{-2, -1\}, \delta = 5$ | 69.78 | 44.71 | 88.99 | 31.25 | 26.20 | 24.56 | 47.58 |

Table 4: **Benchmark Accuracy with Entropy Remasking** We compare the benchmark accuracy on LLADA-8B for our watermark using the recommended hyperparameters (achieving a TPR@1 of 1.0 with the entropy remasking strategy) to that of the unwatermarked model.

| Watermark | MMLU | HeSw | ARC-C | GPQA | MBPP | GSM8K |
|---|---|---|---|---|---|---|
| Unwatermarked | 65.26 | 75.28 | 88.81 | 30.81 | 39.60 | 80.14 |
| $\mathcal{C} = \{-1\}, \delta = 3$ | 55.82 | 72.61 | 88.14 | 29.29 | 35.40 | 75.36 |

the sequence $\omega$. With DLMs, computing the log-likelihood is intractable and only an upper bound can be estimated with Monte Carlo sampling (Nie et al., 2025). Therefore, in Eq. (11), for practical concern, we use $\log p_\theta(\omega)$ as a proxy for the likelihood.

**Experimental Setup**    In practice, we apply GaussMark on LLADA-8B and, because LLADA-8B shares the same architecture as the LLAMA3.2-8B used in Block et al. (2025), we also select as $\theta_d$ the up-projection matrix of the 28th transformer block. To evaluate the GaussMark detectability/quality trade-off, we run the same experiment as in Sec. 4.1.

**Our Approach Outperforms GaussMark**    Fig. 16 shows that our approach provides better detectability than GaussMark given the same impact on quality. Moreover, while the GaussMark overhead at generation time is essentially null, the detection is prohibitively expensive. Indeed, to compute the test statistic in Eq. (11), both a full forward pass and a partial backward pass are needed. Therefore, our watermark also outperforms GaussMark with most practical considerations.

## C.5 ADDITIONAL QUALITY EVALUATION

In this part, we evaluate the influence of our watermark on model utility, measured through LLM benchmark accuracy, and compare it to that of the unwatermarked model.

**Experimental Setup**    We evaluate the benchmark accuracy of LLADA-8B and DREAM-7B, without a watermark, with the baselines, or with our watermark. For both LLADA-8B and DREAM-7B, we use the same watermark hyperparameters as in Table 1. Additionally, for LLADA-8B, we also evaluate the benchmark accuracy with the entropy-remasking strategy to follow the recommended evaluation setting from Nie et al. (2025). For entropy-remasking, the watermark hyperparameters are chosen such that TPR@1 on 300 tokens is around 1.0 (both $\mathcal{C} = \{-1\}, \delta = 3$). For the benchmark, we follow the setup from Nie et al. (2025) and Ye et al. (2025). With LLADA-8B, we use the OpenCompass library (Contributors, 2023), and with DREAM-7B the Evaluation Harness library (Gao et al., 2024). We evaluate the models on MMLU (Hendrycks et al., 2021), HellaSwag (Zellers et al., 2019)

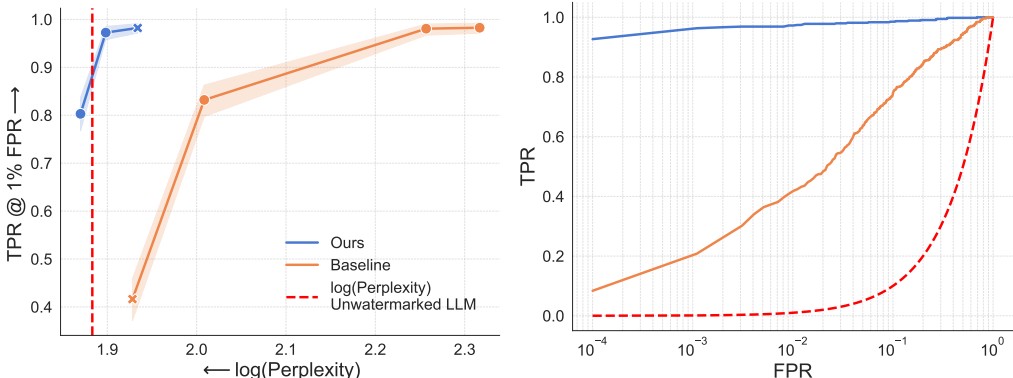

Figure 17: **Detection Performance on Infilling Tasks** (*Left*) We compare the trade-off between watermark detectability (TPR@1) and text quality (log PPL) of our approach and the baseline for different values of the watermark strength parameter $\delta$ and sequences of, on average, 205 tokens. (*Right*) ROC curves of our watermark and the baseline at $\log(PPL) \approx 1.94$. Responses are generated with DREAMON-V0-7B at temperature $0.8$, metrics are computed over 600 samples and we use the previous token as context (i.e., $\mathcal{C} = \{-1\}$). The crosses on the left figure correspond to the same watermark hyperparameters as the right figure.

(HeSw), ARC-C (Clark et al., 2018), GPQA (Rein et al., 2024), MBPP (Austin et al., 2021b), and GSM8K (Cobbe et al., 2021). Lastly, as per Nie et al. (2025) and (Ye et al., 2025), we use low temperature ($T = 0.1$) to compute the benchmark accuracy. In particular, this is a regime in which evaluating a watermark is difficult because the probability distributions are highly spiked and hence, to carry a signal, the watermark is highly distortionary.

**Benchmark With Random Remasking**   Table 3 shows that the impact of the watermark on benchmark accuracy varies greatly between models and benchmarks, with no clear tendencies between models and watermark hyperparameters. On average, we see that although the watermark induces a drop in accuracy, it is relatively small, and most of the time our approach outperforms the baseline. For DREAM-7B in particular, we find that the baseline's performance catastrophically decreases on ARC-C, MBPP and GSM8K, while this is not the case for our watermark.

**Benchmark With Entropy Remasking**   Table 4 shows that, similarly to the random remasking case, the watermark induces a small utility drop on some benchmarks. We find that, as in Table 3, the benchmarks with the highest accuracy drops are MMLU, MBPP, and GSM8K.

## D   WATERMARKING INFILLING TASKS

In this section, we study the effectiveness of our watermark for infilling tasks, rather than instruction tasks as in Sec. 4.

**DreamOn Model**   For this specific task, we consider the DREAMON-V0-7B (Wu et al., 2025) model. DREAMON-V0-7B is an extension of the DREAM-7B model with a novel discrete diffusion algorithm that allows for variable-length generation, an important property for infilling tasks. The vocabulary is augmented with two extra tokens, *expand* and *delete*. During the diffusion process, if the expand token is sampled, it is replaced in the next iterations by two mask tokens. If the delete token is sampled, it is removed from the sequence in the next iterations.

**Experimental Setup**   To evaluate the watermark strength, we use the realnewslike split of the C4 dataset (Raffel et al., 2020), where for each entry we keep a prefix of 100 tokens, mask the next 200 tokens, and leave a suffix of 100 tokens. We then generate infilling with our model for sequences between 100 and 300 tokens long. For each generation, we run the watermark detection and compute the corresponding p-value. In total, we generate 600 sequences. To measure the impact of the watermark on model quality, as in Sec. 4, we measure the text perplexity using QWEN2.5-32B.

For the watermark, we use the same hyperparameters as in Sec. 4: our green list $G$ is generated by sampling i.i.d. Bernoulli random variables that are green with probability $\gamma = 0.25$, we use the SumHash hashing scheme, $\delta$-parameterization, a single iteration for fixed-point convergence, and a top-k of 25. For the generation setting, we set the temperature to $T = 0.8$ (unlike Sec. 4, where the temperature is set to $T = 0.5$), use the entropy-based remasking strategy (see App. C), and allow the model to extend the generated sequences up to 300 tokens.

**Reliable Infilling Watermarking**   Fig. 17 shows the strength of our watermark and the baseline with respect to the text quality. We see that, unlike the ARLM Red-Green watermark, our watermark achieves a strong watermark with virtually no impact on perplexity. These results mean that our watermark can be directly applied to infilling tasks, and even works despite variations in the discrete diffusion process. This is not true for the baseline that particularly struggles in this setting.

## E   LINK TO RED-GREEN ARLM WATERMARKS

In this part, we show that Red-Green Watermarks from Kirchenbauer et al. (2023) are actually a solution to the optimization problem derived in Sec. 3.1, but restricted to the ARLM case.

**Optimization Problem for ARLM**   We keep the same notation as in Sec. 3.1, but adjust it for the ARLM case. Let $\omega \in \Sigma^*$ be a sequence of tokens, $H : \Sigma^* \to \mathcal{H}$ the hash function, and $G \in \{0, 1\}^{\mathcal{H} \times \Sigma}$ the green list matrix. The green ratio function $\hat{\gamma}$ is defined as

$$\hat{\gamma}(\omega) = \frac{1}{|\omega|} \sum_{t=1}^{|\omega|} G_{H(\omega_{<t}), \omega_t} := \frac{1}{|\omega|} \sum_{t=1}^{|\omega|} \hat{\gamma}_t(\omega_{\leq t}). \tag{12}$$

Let $t \in \mathbb{N}$. Given $\omega_{<t}$, the ARLM returns a next-token probability distribution $p_t \in \Delta(\Sigma)$. The goal of the watermarking algorithm is to distort the distribution $p_t$ into a distribution $q_t$ that maximizes the expected green ratio, which we formalize as

$$\text{maximize}_{q_t \in \Delta(\Sigma)} \; \mathbb{E}_{\Omega \sim q_t}[\hat{\gamma}_t(\omega_{<t} \circ \Omega)], \tag{13}$$
$$\text{subject to } \mathrm{KL}(q_t, p_t) \leq \varepsilon$$

where $\circ$ is the concatenation operator. Given that $H(\omega_{<t})$ is a constant, unrolling the expectation is significantly easier than in Sec. 3.1. We simply get

$$\forall q \in \Delta(\Sigma), \mathbb{E}_{\Omega \sim q}[\hat{\gamma}_t(\omega_{<t} \circ \Omega)] = \sum_{u \in \Sigma} G_{H(\omega_{<t}), u} q(u) =: J_t(q). \tag{14}$$

**Link to Red-Green ARLM Watermarks**   Similarly to Theorem 3.1, there exists a unique $\delta > 0$ such that the optimal solution is given by

$$\exists c \in \mathbb{R}, \forall u \in \Sigma, \log q_t^*(u) = \log p_t(u) + \delta G(H(\omega_{<t}), u) + c. \tag{15}$$

This is exactly the formulation of the Red-Green watermark from Kirchenbauer et al. (2023), where only the logits of the green tokens are boosted by a constant $\delta$.

## F   EXPERIMENTING WITH DIFFERENT GREEN LIST DISTRIBUTIONS

In this section, we present the technical details of the Red-Green watermark scheme. We focus on how the choice of a correlated green list—that is, fixing its size to exactly $\gamma|\Sigma|$—affects detection. Prior work enforced this constraint by using correlated Bernoulli variables, whereas we instead draw the green list with i.i.d. Bernoulli variables. This change, originally introduced for technical reasons, enables an exact test for watermark detection, while previous approaches rely only on approximations.

**Green List Distribution**   In the original work by Kirchenbauer et al. (2023), they suggest that, given a fixed context, the Red-Green list should partition the vocabulary $\Sigma$ with exactly $\gamma|\Sigma|$ green tokens

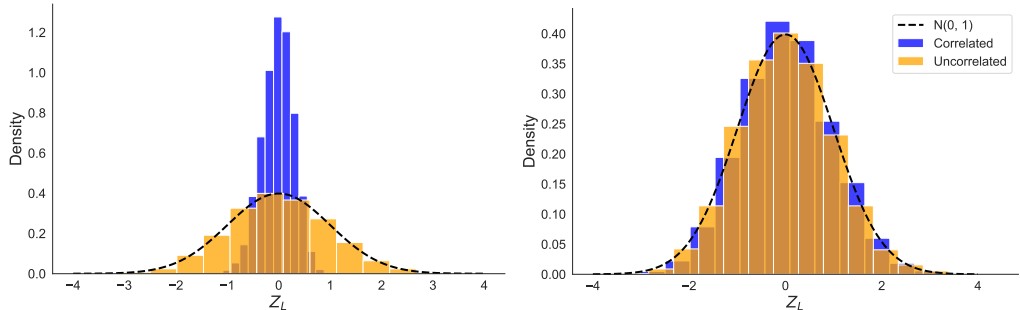

Figure 18: **Z-score Simulation** Simulation of $Z_L$ using a uniform token sequence in $\Sigma$, with either the correlated or independent green list. The black dotted line is the standard normal pdf. (*Left*) We use $|\Sigma| = 15$ and $L = 200$. (*Right*) We use $|\Sigma| = 50$ and $L = 200$.

and the rest red tokens. Let $t \in \Sigma$, $s \in \mathcal{H}$ be a hash, and $G_{s,t}$ the random variable corresponding to the color of token $t$ with the context hash being $s$. This means that ($\perp$ means independent),

$$\forall s \in \mathcal{H}, \forall t \in \Sigma, G_{s,t} \sim \mathcal{B}(\gamma), \tag{16}$$

$$\forall t \neq t' \in \Sigma, \text{Cov}(G_{s,t}, G_{s,t'}) = -\frac{\gamma(1-\gamma)}{|\Sigma|-1}, \tag{17}$$

$$\forall s \neq s' \in \mathcal{H}, \forall t, t' \in \Sigma, G_{s,t} \perp G_{s',t'}. \tag{18}$$

On the contrary, in this work, we instead use i.i.d. Bernoulli for the green list, so we simply have

$$\forall s \in \mathcal{H}, \forall t \in \Sigma, G_{s,t} \sim \mathcal{B}(\gamma) \tag{19}$$

$$\forall s, s' \in \mathcal{H}, \forall t \neq t' \in \Sigma, G_{s,t} \perp G_{s',t'} \tag{20}$$

$$\forall s \neq s' \in \mathcal{H}, \forall t \in \Sigma, G_{s,t} \perp G_{s',t}. \tag{21}$$

**Z-score Detector** In Kirchenbauer et al. (2023), given a sequence of tokens $t_1, \ldots, t_L$ and a corresponding hash sequence $s_1, \ldots, s_L$ such that there is no repetition of tuples $(t_i, s_i)$, they suggest using as a detector

$$Z_L = \frac{1}{\sqrt{\gamma(1-\gamma)L}} \left( \sum_{i=1}^{L} G_{s_i,t_i} - \gamma L \right), \tag{22}$$

and assume asymptotic standard normality, i.e., $Z_L \to^{\mathcal{D}} \mathcal{N}(0, 1)$. However, because of potential hash repetition, the variance normalization may be inaccurate. Indeed, let, for all $s \in \mathcal{H}$, $N_s := \{k \in [1, \ldots, L] : s_k = s\}$ which corresponds to the indices $k$ where the hash is equal to $s$. Hence, $|N_s|$ counts the repetitions of the hash $s$ in the sequence. Then, we have

$$\sigma^2 := \text{Var}\left( \sum_{i=1}^{L} G_{s_i,t_i} \right) = \gamma(1-\gamma)L \left[ 1 - \frac{1}{L(|\Sigma|-1)} \sum_{s \in \mathcal{H}} (|N_s|^2 - |N_s|) \right]. \tag{23}$$

This means that a corrected Z-score would be

$$Z_L = \frac{1}{\sigma} \left( \sum_{i=1}^{L} G_{s_i,t_i} - \gamma L \right). \tag{24}$$

In our work with an i.i.d. Bernoulli green list, we can use for detection Eq. (22) directly without modifying the variance.

In Fig. 18, we simulate $Z_L$ with both the correlated and i.i.d. green lists using $\mathcal{H} = \Sigma$, with $|\Sigma| \in \{15, 20\}$ and $L = 100$. When $\Sigma$ is small and the sequence is long (left), we notice that in the correlated case Eq. (22) deviates from the standard normal distribution, whereas this is not the case for the independent one. However, when $\Sigma$ is large enough compared to the sequence length (right), there is almost no deviation from the normal distribution—which is expected given the variance formulation in Eq. (23). Therefore, in most practical cases, the slight deviation from standard normality has almost no impact on the reported p-values by the watermark detection as we operate in the $\Sigma >> L$ regime. But this derivation and these experiments justify our claim in Sec. 4 that using i.i.d. Bernoulli with the suggested detection is more principled.

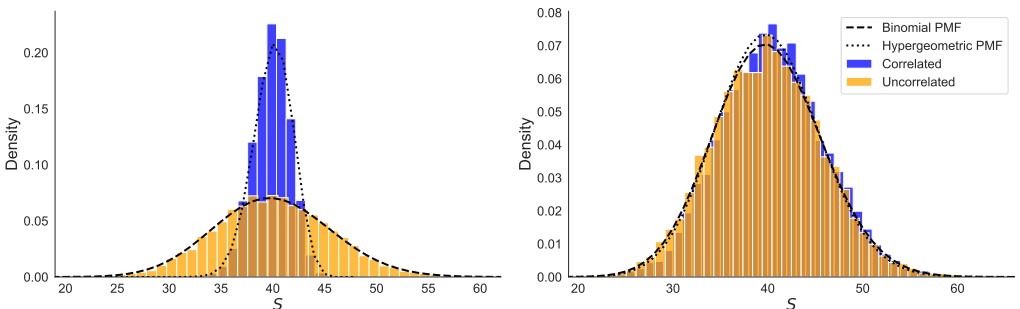

Figure 19: **Binomial Score Simulation** Simulation of $S$ using a uniform token sequence in $\Sigma$, with either the correlated or independent green list. The black dotted line is the standard normal pdf. (*Left*) We use $|\Sigma| = 15$ and $L = 200$. (*Right*) We use $|\Sigma| = 50$ and $L = 200$.

**Binomial Detector**   As explained in Fernandez et al. (2023a), using a z-score for detection is imprecise and leads to higher empirical FPR than expected. Therefore, in this work, we use a binomial test instead. Yet, similarly as with the z-score detection, the binomial test is rigorously valid only when using i.i.d. Bernoulli green list and not the correlated one.

Let $t_1, \ldots, t_L$ be a sequence of tokens and $s_1, \ldots, s_L$ the corresponding sequence of hashes such that there is no repetition of tuples $(t_i, s_i)$. The binomial detector is based on

$$S = \sum_{i=1}^{L} G_{s_i, t_i}. \tag{25}$$

In the i.i.d. case, we know that for all $i \neq j$, $G_{s_i, t_i} \perp G_{s_j, t_j}$ and $G_{s_i, t_i} \sim \mathcal{B}(\gamma)$. Hence, $S$ is the sum of $L$ i.i.d. Bernoulli random variables: it follows exactly a binomial distribution of parameter $(L, \gamma)$.

In the correlated case, we introduce for all $s \in \mathcal{H}$, $N_s := \{k \in [1, \ldots, L] : s_k = s\}$ which corresponds to the indices $k$ where the hash is equal to $s$. Then, we have

$$S = \sum_{s \in \mathcal{H}} \left( \sum_{i \in N_s} G_{s, t_i} \right) := \sum_{s \in \mathcal{H}} X_{N_s}. \tag{26}$$

We have that $X_{N_s} | N_s$ follows a hypergeometric distribution of parameter $(|\Sigma|, \gamma|\Sigma|, N_s)$. This means that the distribution of $S$ depends on the distribution of $N_s$ which itself depends on the distribution of the LLM. Hence, we can't provide a closed-form distribution for $S$. Yet, if we add the assumption that the tuples $(t_i, s_i)$ are sampled uniformly without replacement, we show that $S$ follows a hypergeometric distribution of parameter $(|\mathcal{H}||\Sigma|, \gamma|\mathcal{H}||\Sigma|, L)$.

*Proof.* Let $\mathcal{I} = \{i \in \{0, \ldots, L\}^{|\mathcal{H}|}, \sum_{s \in \mathcal{H}} i_s = L\}$. First, we have

$$\forall i \in \mathcal{I}, \mathbb{P}[\forall s \in \mathcal{H}, N_s = i_s] = \frac{\prod_{s=1}^{|\mathcal{H}|} \binom{|\Sigma|}{i_s}}{\binom{|\mathcal{H}||\Sigma|}{L}}. \tag{27}$$

We now develop $S$ by conditioning on $N_s$, thus we have for all $k \in \{0, \ldots, L\}$,

$$\mathbb{P}[S = k] = \sum_{i \in \mathcal{I}} \mathbb{P}[\forall s, N_s = i_s] \sum_{\substack{k_1 + \ldots + k_{|\mathcal{H}|} = k \\ 0 \leq k_s \leq i_s}} \prod_{s \in \mathcal{H}} \mathbb{P}[X_{N_s} = k_s | N_s = i_s] \tag{28}$$

$$= \frac{1}{\binom{|\mathcal{H}||\Sigma|}{L}} \sum_{i \in \mathcal{I}} \sum_{\substack{k_1 + \ldots + k_{|\mathcal{H}|} = k \\ k_s \geq 0}} \prod_{s \in \mathcal{H}} \binom{\gamma|\Sigma|}{k_s} \binom{|\Sigma| - \gamma|\Sigma|}{i_s - k_s}. \tag{29}$$

Using Vandermonde's identity we have that

$$\sum_{i \in \mathcal{I}} \prod_{s \in \mathcal{H}} \binom{|\Sigma| - \gamma|\Sigma|}{i_s - k_s} = \binom{|\mathcal{H}||\Sigma|(1 - \gamma)}{L - k}, \tag{30}$$

$$\sum_{\substack{k_1 + \ldots + k_{|\mathcal{H}|} = k \\ k_s \geq 0}} \prod_{s \in \mathcal{H}} \binom{\gamma|\Sigma|}{k_s} = \binom{\gamma|\mathcal{H}||\Sigma|}{k}. \tag{31}$$

Hence if we combine the last three equations we have that

$$\mathbb{P}[S = k] = \frac{1}{\binom{|\mathcal{H}||\Sigma|}{L}} \binom{|\mathcal{H}||\Sigma|(1 - \gamma)}{L - k} \binom{\gamma|\mathcal{H}||\Sigma|}{k}, \tag{32}$$

which is exactly the pmf of the hypergeometric distribution of parameter $(|\mathcal{H}||\Sigma|, \gamma|\mathcal{H}||\Sigma|, L)$. $\square$

In Fig. 19, we simulate $S$ with both the correlated and i.i.d. green lists using $\mathcal{H} = \Sigma$, with $|\Sigma| \in \{15, 20\}$ and $L = 100$. We see, in the left figure, that when $\Sigma$ is small (i.e., $|\mathcal{H}||\Sigma| \approx L$) we are in a regime where the hypergeometric and binomial distributions are very different, and the two histograms differ significantly. However, in the right figure, where $|\mathcal{H}||\Sigma| >> L$, we see that the hypergeometric and binomial distributions are similar, and so are the histograms. Moreover, in both figures, we see that the empirical histograms match the corresponding theoretical distributions. This experimental validation supports our derivation and the rigor of using a binomial test to compute watermark detection p-values when using an i.i.d. Bernoulli green list.

## G  ADDITIONAL IMPLEMENTATION DETAILS

### G.1  OUR HASH IMPLEMENTATION

---
**Algorithm 3** $\varepsilon$-parameterization bisection
---
**Require:** DLM probabilities $p_t$, $\alpha_t$ derivative of the energy function, target $\varepsilon$.
1:  $\delta_{\text{low}} \leftarrow 0$, $\delta_{\text{high}} \leftarrow 2\sqrt{\frac{2\varepsilon}{\text{Var}_{p_t}(\alpha)}}$         $\triangleright$ Bisection Bounds
2:  $\delta_{\text{high}} \leftarrow \min(\delta_{\text{high}}, 10^4)$
3:  **for** $i = 1$ to 16 **do**         $\triangleright$ 16 iterations means a resolution of 0.15
4:       $\delta_{\text{mid}} \leftarrow \frac{1}{2}(\delta_{\text{low}} + \delta_{\text{high}})$
5:       $q \leftarrow p_t \exp(\alpha_t \delta_{\text{mid}})$
6:       $k \leftarrow \sum q \log \frac{q}{p_t}$         $\triangleright$ KL$(q \| p_t)$
7:       **if** $(k > \varepsilon)$ **then** $\delta_{\text{high}} \leftarrow \delta_{\text{mid}}$ **else** $\delta_{\text{low}} \leftarrow \delta_{\text{mid}}$
8:  **end for**
9:  **return** $\delta_{\text{low}}$
---

In this part, we show that our instantiations of SumHash and MinHash are functionally equivalent to those of Kirchenbauer et al. (2024).

**SumHash** Our SumHash implementation corresponds to the Additive implementation from Kirchenbauer et al. (2024). Indeed, given a sequence of tokens $\omega \in \Sigma^*$, Additive is defined by

$$H^{Additive}(\omega)_t = P\left(s \times \sum_{i \in \mathcal{C}} \omega_{t+i}\right), \tag{33}$$

where $s \in \mathbb{N}$ is a seed and $P : \mathbb{N} \rightarrow \mathcal{H}$ a PRF. Hence, $H^{Additive}$ maps every unique sum of token IDs from the context to a unique value, which is then used to sample the green list. In our case, $H^{SumHash}$ also maps every unique sum of token IDs from the context to a unique value, which is used to select a row, sampled randomly, from the green list. In the end, the results are similar: every unique sum of token IDs gets associated with a random green list.

**MinHash** For MinHash, the reasoning is similar: both the approach from Kirchenbauer et al. (2024) and our approach associate the minimum token id of the context (up to a permutation) with a unique green list. The main difference is that we represent our green list as a "pre-generated" matrix and thus do not need to further use a PRF function.

### G.2 $\varepsilon$-PARAMETERIZATION IMPLEMENTATION

In this part we explain the $\varepsilon$-parameterization implementation described in Algorithm 3.

For every token position $t$, given the DLM probabilities $p_t$, the derivative of the energy function $\alpha_t$, and the target $\varepsilon$, the goal is to find the unique $\delta^*$ such that $KL(q_t(\delta)||p_t) = \varepsilon := KL(\delta)$. We find in Theorem G.1 that $KL(\delta)$ is strictly increasing in $\delta$. Hence, for the bisection lower bound we use $\delta_{\text{low}} = 0$ (line 1), and for the upper bound, using the approximation from Theorem G.1, we use $\delta_{\text{high}} = 2\sqrt{\frac{2\varepsilon}{\text{Var}_{p_t}(\alpha_t)}}$ (line 1). For practical purposes, we cap the upper bound at $10^4$ (line 2). This means that in some cases we might reach a lower KL-divergence than $\varepsilon$ but will never exceed it. We then compute the midpoint (line 4) and compute the KL-divergence at the midpoint (line 6). If it is smaller than $\varepsilon$, it means that $\delta^*$ is higher than the midpoint and we increase the lower bound; otherwise, we decrease the upper bound (line 7). We return the lower bound (line 8) at the end of the iteration to never exceed the KL-threshold. In practice, we find that 16 iterations are enough to systematically obtain a KL-divergence nearly equal to $\varepsilon$.

The complexity of each iteration of Algorithm 3 is $O(|\Sigma|)$ (for computing the KL divergence). If using top-$k$, it reduces to $O(1)$. Because the number of iterations is constant, the complexity is $O(1)$.

**Theorem G.1.** *The function $KL(\delta)$ is strictly increasing in $\delta$. Moreover, to second order in $\delta$,*

$$KL(\delta) = \frac{1}{2}Var_{p_t}(\alpha_t)\delta^2 + O(\delta^3). \tag{34}$$

*Proof.* Let

$$Z(\delta) = \sum_{u \in \Sigma} p_t(u) \exp(\delta\alpha_t(u)). \tag{35}$$

Note that

$$KL(\delta) = \sum_{u \in \Sigma} p_t(u) \exp(\delta\alpha_t(u))[\delta\alpha_t(u) - \log Z(\delta)]. \tag{36}$$

By taking the derivative with respect to $\delta$, we find that

$$KL'(\delta) = \delta \, \text{Var}_{p_t \exp(\delta\alpha_t)}[\alpha_t] > 0. \tag{37}$$

Hence, $KL(\delta)$ is strictly increasing in $\delta$. By Taylor expansion, we have that

$$\log Z(\delta) = \delta\mathbb{E}_{p_t \exp(\delta\alpha_t)}[\alpha_t] + \frac{1}{2}\delta^2\text{Var}_{p_t}[\alpha_t] + O(\delta^3). \tag{38}$$

Hence,

$$KL(\delta) = \frac{1}{2}\text{Var}_{p_t}(\alpha_t)\delta^2 + O(\delta^3). \tag{39}$$

$\square$

## H ON THE NO SELF-HASH ASSUMPTION

In this section, we discuss the no self-hash assumption introduced in Sec. 3, and evaluate a simple self-hashing scheme (corresponding to SumHash with $\mathcal{C} = \{0\}$).

**Derivation Without the Assumption** We recall the notation from Sec. 3. Let $p \in \Delta(\Sigma)^L$ be the factorized probability distribution over the set of sequences, let $\Omega \sim p$ be a random sequence, and $H_t : \Sigma^L \to \mathcal{H}$ the function that returns the hash of the context at position $t$ (without the no self-hashing assumption). We denote by $\mathbb{P}_q$ the probability under the distribution $q \in \Delta(\Sigma)^L$. Then, Eq. (2) becomes

$$\forall q \in \Delta(\Sigma)^L, \mathbb{E}_{\Omega \sim q}[\hat{\gamma}(\Omega)] = \frac{1}{L}\sum_{t=1}^{L}\sum_{h \in \mathcal{H}}\sum_{u \in \Sigma} G_{h,u}\mathbb{P}_q(H_t = h, \Omega_t = u). \tag{40}$$

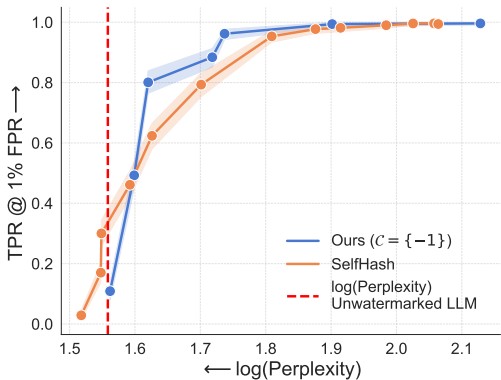

Figure 20: **Evaluation of Our SelfHash Scheme** We study the trade-off between detectability (TPR@1) and text quality (log PPL) of our approach using the SelfHash scheme; and compare it to SumHash with $\mathcal{C} = \{-1\}$. Responses are generated with LLADA-8B at temperature $0.7$ and metrics averaged over $600$ samples.

With the no self-hash assumption, we could further simplify

$$\forall q \in \Delta(\Sigma)^L, \forall t, \forall h \in \mathcal{H}, \forall u \in \Sigma, \mathbb{P}_q(H_t = h, \Omega_t = u) = \mathbb{P}_q(H_t = h)\mathbb{P}_q(\Omega_t = u). \tag{41}$$

Yet, without the no self-hash assumption, by letting

$$\forall q \in \Delta(\Sigma)^L, J(q) = \frac{1}{L}\sum_{t=1}^{L}\sum_{h \in \mathcal{H}}\sum_{u \in \Sigma} G_{h,u}\mathbb{P}_q(H_t = h, \Omega_t = u), \tag{42}$$

then Algorithm 2 still holds. The issue is simply how to compute $J(q)$ in practice.

**A Simple Self-Hash Hashing Scheme** We consider a simple self-hash scheme, $\forall t, \forall \omega \in \Sigma^L, H_t(\omega) = \omega_t$. This scheme corresponds to SumHash with $\mathcal{C} = \{0\}$. With this scheme, we have

$$\forall q \in \Delta(\Sigma)^L, \forall t, \forall h \in \mathcal{H}, \forall u \in \Sigma, \mathbb{P}_q(H_t = h, \Omega_t = u) = 1\{h = u\}q_t(u). \tag{43}$$

Hence, we have

$$\forall q \in \Delta(\Sigma)^L, J(q) = \frac{1}{L}\sum_{t=1}^{L}\mathrm{diag}(G) \cdot q_t. \tag{44}$$

Given this $J$, we can simply apply Algorithm 2 to watermark a DLM with SelfHash. Note that the underlying scheme is very simple; we have

$$\forall t, q_t^* \propto p_t \exp(\delta_t \mathrm{diag}(G)). \tag{45}$$

This means that the watermark simply consists of increasing the probabilities of tokens whose hash makes them green.

Fig. 20 shows the detectability/quality trade-off of our watermark using the SelfHash scheme described above. We find that, compared to SumHash with $\mathcal{C} = \{-1\}$, this trade-off is relatively similar.

# I  RESOURCES

## I.1  COMPUTATIONAL RESOURCES

All experiments presented in this work were conducted on either a single H100 (24 vCPU) GPU node with 80GB of memory (hosted by Lambda Labs) or a single A100 (24 vCPU) with 40GB of memory.

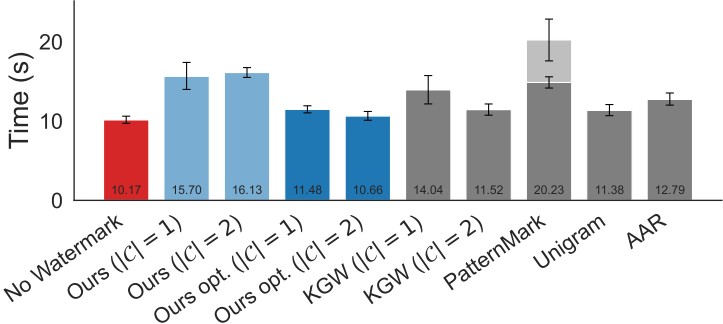

Figure 21: **Implementation Speed of Our Watermark** We compare the average time, over $5 \times 300$ repetitions, to generate a 300-token-long sequence with 300 diffusion steps on LLADA-8B with different watermarks applied and the corresponding average detection time (in lighter colors). *This plot shows implementation speed on a given hardware, and variations due to the hardware (e.g., temperature) are not properly controlled.* For most watermarks, the detection time is negligible compared to the generation time and thus not visible. We highlight our methods in light blue and an optimized implementation in blue, and the base model without watermark in red.

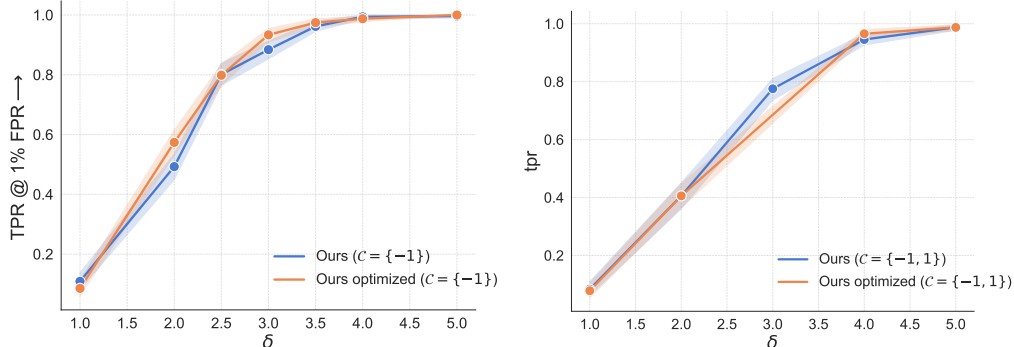

Figure 22: **Comparison of the Main and Optimized Watermark Implementation** We compare the watermark detectability (TPR@1) with respect to the watermark strength parameter $\delta$ of our main implementation and the optimized one, averaged over 600 sequences of on average 275 tokens. We find that both implementations are equivalent in terms of watermark detectability.

**Runtime Analysis** Fig. 21 shows the average time, over $5 \times 300$ repetitions, in seconds to generate a 300-token-long sequence with 300 diffusion steps on LLADA-8B with different watermarks. The numbers only reflect the average speed required to run the experiments given specific hardware under specific conditions (e.g., temperature). Importantly, across watermarks, we could not control such external factors, which may significantly impact the observed runtime. In particular, for each watermark we run the experiment (300 repetitions) 5 times non-consecutively to account for such factors. KGW corresponds to the baseline presented in Sec. 4, PatternMark and Unigram are prior order-agnostic watermarks presented in detail in Sec. 4.4, and AAR is a baseline adaptation of the watermark from Aaronson (2023) introduced in App. C. For our watermark, we use the default hyperparameters from Sec. 4. We observe that the overhead introduced by our watermark is minimal (only +9ms/token with $\mathcal{C} = \{-1\}$), even though our implementation is not optimized for speed. With an optimized implementation (Ours opt.), the speed becomes similar to that of the baselines. Importantly, unlike PatternMark, our detection time is negligible. Because detection is applied indiscriminately to any text, it is crucial for it to be fast.

**Optimized Implementation** While our main contribution is the watermark algorithm Algorithm 2, we tested whether we could implement it in a fast and efficient manner. Unlike the main implementation used for all our experiments (Sec. 4), which is very general and can be easily adapted to new DLM implementations, the fast implementation is designed specifically for the LLADA-8B implementation with random remasking. Specifically, we tightly integrated the watermark into the remasking

procedure. In the main implementation, we compute the watermarked distribution for every token position at every step (even for already unmasked tokens). With the optimized version, we compute the watermarked distribution only for the tokens to be unmasked. Hence, both implementations are functionally equivalent. To experimentally validate their equivalence, we run the same experiments as in Sec. 4.1 with the fast implementation and compare them to the main implementation. Fig. 22 shows that both implementations yield the same results, validating their equivalence.

## I.2 LLM USAGE

In this work, we use LLMs as coding assistants and to make minor grammatical and stylistic changes to the paper. Importantly, no content in this paper was generated by LLMs, except for the watermarked text examples in App. K.2.

## J PROOFS

### J.1 SOLUTION OF THE OPTIMIZATION PROBLEM

We recall Theorem 3.1:

**Theorem 3.1.** *Given $p \in \Delta(\Sigma)^L$ and $J$ defined in Eq. (2), there exists $\delta \in \mathbb{R}^L$ such that*

$$\forall t \in [1, \ldots, L], q_t^* \propto p_t \exp(\delta_t \alpha_t(q^*)) \tag{4}$$

*with $\alpha_t(q) = \nabla_{q_t} J(q)$. Moreover, for all $t \in [1, \ldots, L]$, $\delta_t$ is the unique solution to $KL(q_t^*, p_t) = \varepsilon$.*

*Proof.* We first recall the optimization problem from Eq. (1),

$$\text{maximize}_{q \in \Delta(\Sigma)^L} \ J(q) = \sum_{t=1}^{L} h_t(q)^T \cdot G \cdot q_t \tag{46}$$

$$\text{subject to } \forall t \in [1, \ldots, L], \text{KL}(q_t, p_t) \leq \varepsilon \tag{47}$$

where for all $t \in [1, \ldots, L]$, $h_t \in \Delta(H)$, $p_t \in \Delta(\Sigma)$ and $G \in \{0, 1\}^{|H| \times |\Sigma|}$, and $H, \Sigma$ are finite non-empty sets. We recall the definition of the KL-divergence,

$$\forall t \in [1, \ldots, L], KL(q_t, p_t) = \sum_{u \in \Sigma} q_t(u)(\log(q_t(u)) - \log(p_t(u))). \tag{48}$$

Let $\lambda \in \mathbb{R}^L, \mu \in \mathbb{R}^L$, we introduce the Lagrangian

$$\mathcal{L}(q, \lambda, \mu) = J(q) - \sum_{t=1}^{L} \lambda_t(KL(q_t, p_t) - \varepsilon) - \sum_{t=1}^{L} \mu_t(\sum_{u \in \Sigma} q_t(u) - 1). \tag{49}$$

Because $J$ is continuous and the set of constraints is compact, $J$ attains a maximum $J^*$ that is reached on the set of constraints. Furthermore, Slater's conditions are verified because $p$ is within the set of constraints and satisfies for all $t$, $KL(p_t, p_t) \leq \varepsilon$. Let $q^*$ be a point that reaches $J^*$. We know that $q^*$ satisfies the Karush-Kuhn-Tucker (KKT) condition. Hence,

$$\forall t \in [1, \ldots, L], \forall u \in \Sigma, \alpha_t(q^*)(u) - \lambda_t \left(1 + \log(\frac{q_t^*(u)}{p_t(u)})\right) - \mu_t = 0 \tag{50}$$

where $\alpha_t(q) = \nabla_{q_t} J(q)$. Thus, with $\delta = 1/\lambda$, we find Eq. (4),

$$\forall t \in [1, \ldots, L], q_t^* \propto p_t \exp(\delta_t \alpha_t(q^*)). \tag{51}$$

Furthermore, we know thanks to the KKT condition that assuming the constraint is active

$$\forall t \in [1, \ldots, L], KL(q_t^*, p_t) = \varepsilon. \tag{52}$$

Because for all $t \in [1, \ldots, L]$ the KL-divergence is monotone with respect to $\delta_t$, this guarantees uniqueness of $\delta_t$, the solution of Eq. (52).

$\square$

## J.2 DEFINITION OF EXPECTATION BOOST AND PREDICTIVE BIAS TERMS

In Sec. 3.3, we show that, in the case of SumHash with $\mathcal{C} = \{-1\}$, our watermark algorithm can be split into two terms: the expectation boost term, similar to the Red-Green watermark boost, and the predictive bias term that favors sampling a token whose hash makes the most likely next tokens more likely to be green. In this part, we derive the computations to define the expectation boost and predictive bias terms in the general setting.

We recall that $h_t : \Delta(\Sigma)^L \to \Delta(\mathcal{H})$ is the function that maps a factorized token probability vector to a corresponding hash probability vector at position $t$, and that $p \in \Delta(\Sigma)^L$ is the factorized probability vector over the sequence given by our DLM. Lastly, we introduce the factorized hash probability distribution $h \in \Delta(\mathcal{H})^L$ defined as $h = (h_1, \dots, h_L)$.

With Theorem 3.1, we have that, at the first order

$$\forall t \in [1, \dots, L], q_t^* \propto p_t \exp(\delta \alpha_t(p)). \tag{53}$$

Yet, if we distribute the derivative in $\alpha_t = \nabla_{p_t} J(p)$ (with $J$ defined in Eq. (2)), we get that

$$\alpha_t(p) = G^\top h_t(p) + G \nabla_{p_t} h(p). \tag{54}$$

$G^\top h_t(p)$ corresponds to the expectation boost term: it can be interpreted as applying the Red-Green watermark boost in expectation over the distribution of the context hashes. Therefore, $G \nabla_{p_t} h(p)$ is the predictive bias term, but it is not as easily interpretable without explicitly deriving the gradient.

Under the self-hashing assumption, i.e., $0 \notin \mathcal{C}$, we have that $\nabla_{p_t} h_t(p) = 0$. We also assume that the hash distribution $h_t$ is given by

$$\forall h \in \mathcal{H}, \ h_t(h) = \sum_{\substack{u_1, \dots, u_L \in \Sigma^L \\ H_t(u_1, \dots, u_L) = h}} p_1(u_1) \dots p_L(u_L) \tag{55}$$

Given these assumption, we have that

$$G \nabla_{p_t} h(p) = \sum_{s \neq t \in [1, \dots, L]} (\nabla_{p_t} h_s(p))^\top G p_s. \tag{56}$$

Here, the assumption $\nabla_{p_t} h_t(p) = 0$ removed the self-feedback term $(\nabla_{p_t} h_t(p))^\top G p_t$. We have, given $u \in \Sigma$ and $h \in \mathcal{H}$, and for $s \neq t \in [1, \dots, L]$,

$$\frac{\partial h_s(p)_h}{\partial p_t(u)} = \sum_{u_{-t} \in \Sigma^{L-1}} \mathbf{1}\{H_s(u, u_{-t}) = h\} \prod_{i \neq t} p_i(u_i) =: \mathbb{P}[H_s(\Omega) = h | \Omega_t = u], \tag{57}$$

with $\Omega \sim p$ the random variable representing sequences of tokens distributed according to the factorized probability distribution $p$. Hence, by distributing the sums we get that for all $t \in [1, \dots, L]$ and $u \in \Sigma$,

$$\alpha_t(p)_u = \underbrace{\sum_{h \in \mathcal{H}} G_{h,u} h_t(p)_h}_{\text{expectation boost}} + \underbrace{\sum_{s \neq t} \sum_{h \in \mathcal{H}} \mathbb{P}[H_s(\Omega) = h | \Omega_t = u](G p_s)_h}_{\text{predictive bias}}. \tag{58}$$

We see here that the predictive bias term promotes tokens $u$ at position $t$ whose induced hash distribution on positions $s \neq t$ makes the most probable tokens at $s$ green.

## K EXTENDED RESULTS

In this section, we provide extended results for the evaluation from Sec. 4. Specifically, we include the ROC curves for most of the experiments, as well as some text examples of unwatermarked and watermarked text.

### K.1 ROC CURVES

In this part, we show the ROC curves for the experiment from Sec. 4. Such curves may guide practitioners in deciding which hyperparameters best suit their needs.

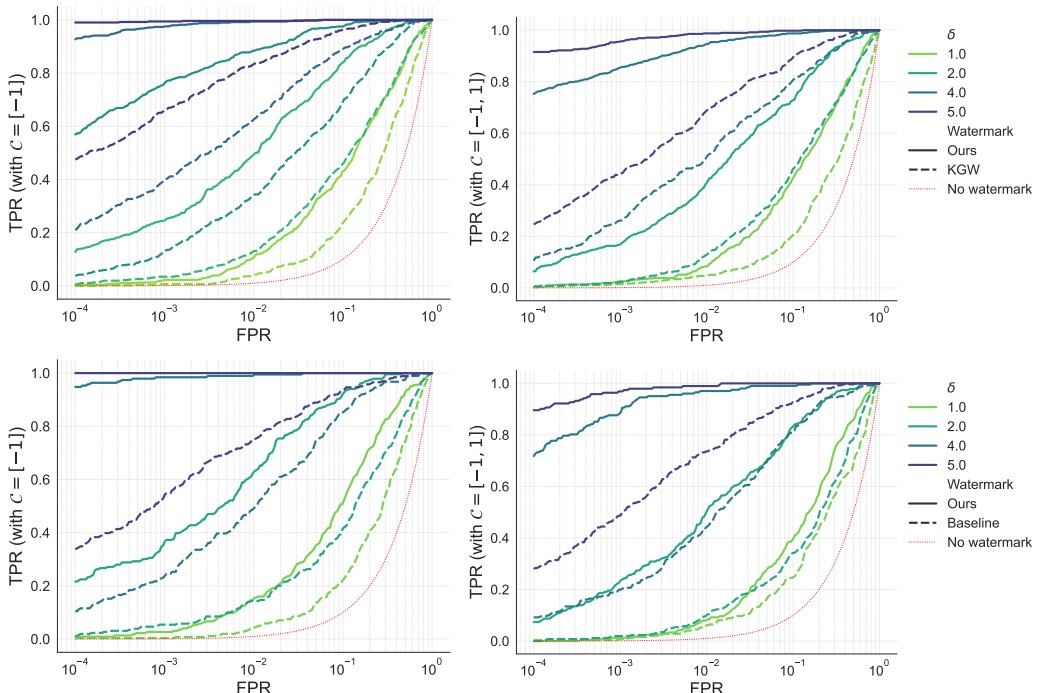

Figure 23: **Watermark Performance** ROC curves (log scaled) of KGW and our watermark for both LLADA-8B (top) and DREAM-7B (bottom), and different values of $\delta$ using $\mathcal{C} = \{-1\}$ (left) or $\mathcal{C} = \{-1, 1\}$ (right).

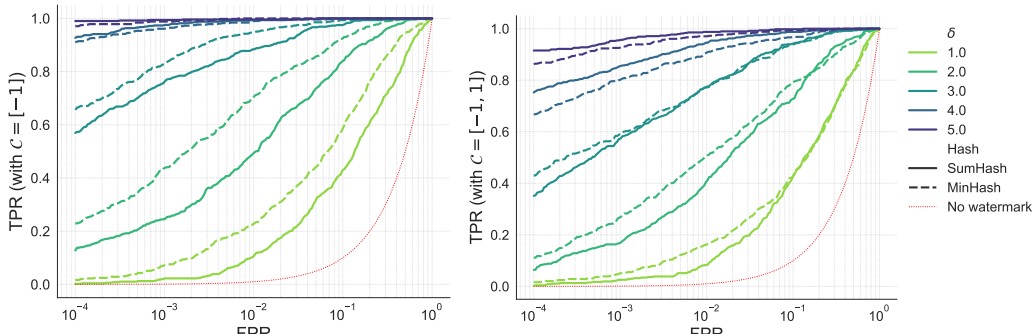

Figure 24: **Ablation on the Hashing Scheme** ROC curves (log scaled) of our watermark with different hashing schemes for LLADA-8B, and different values of $\delta$ using $\mathcal{C} = \{-1\}$ (left) or $\mathcal{C} = \{-1, 1\}$ (right).

**Main experiments** In Fig. 23, we show the ROC curves from the experiments in Sec. 4. We observe that for $\delta \geq 4$, we achieve a TPR@1 of around 99% with our watermark, while the baseline remains at most 80%. Hence, for a strong watermark at low temperature (here $T = 0.5$) and in the instruction setting, we recommend that practitioners use $\delta = 4$.

**Hashing Scheme** Fig. 24 shows the ROC curves for both SumHash and MinHash. We see that the watermark strength is not significantly influenced by the choice of hash. In fact, the choice of hash should be dictated by other concerns such as watermark security or robustness, as explored in prior works (Jovanović et al., 2024; Kirchenbauer et al., 2024).

**Fixed-point Iteration** Fig. 25 shows the ROC curves with LLADA-8B for different numbers of fixed-point iterations. The conclusion is similar to Sec. 4.3: increasing the number of iterations only

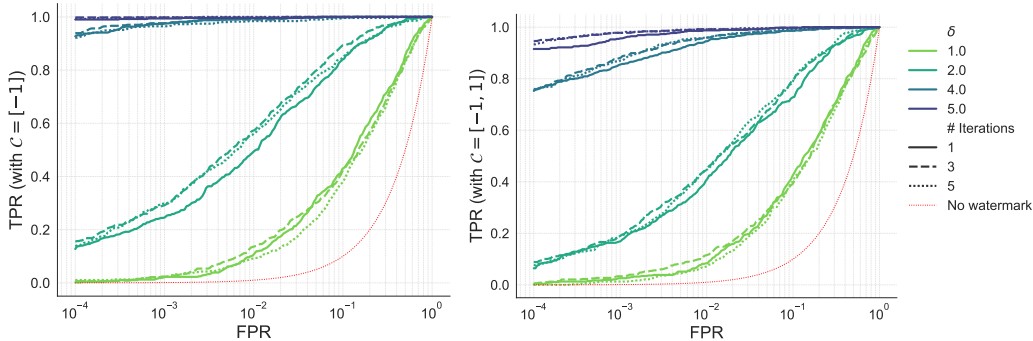

Figure 25: **Ablation on the Fixed-Point Iteration** ROC curves (log scaled) of our watermark with different number of fixed-point iterations for LLADA-8B, and different values of $\delta$ using $\mathcal{C} = \{-1\}$ (left) or $\mathcal{C} = \{-1, 1\}$ (right).

marginally increases the watermark strength, and the increase plateaus quickly. Thus, we advise practitioners to use only a single iteration.

### K.2 TEXT SAMPLES

In this part, we show examples of prompts and model answers for both unwatermarked text and text generated using our watermark (with $\delta = 4$ and $\gamma = 0.25$). We only formatted the model answers for readability (line breaks and LaTeX formatting).

---

**User Prompt**

Please tell me about the romantic relationship between Julius Caesar and Cleopatra

---

**Llada-8B (No Watermark)**

Julius Caesar and Cleopatra were both historical figures, but they did not have a romantic relationship with each other. Julius Caesar was a famous Roman general, and Cleopatra was a powerful queen of Egypt. However, they did have a close relationship that is often interpreted as romantic.

According to history, Caesar met Cleopatra while visiting her kingdom during his campaign in 48 BCE. He was impressed by her beauty, and they began a close friendship. However, it's unlikely that Caesar and Cleopatra had a romantic relationship during this early period.

In 43 BCE, Julius Caesar and Cleopatra allegedly engaged in a romantic relationship after Caesar was assassinated by a mob in Rome. Caesar's death led to a brief civil war, and Cleopatra traveled to Rome in support of her brother, Antony, who wanted to secure Caesar's legacy and restore Rome.

According to historical accounts, Cleopatra's presence in Rome led to two children with Antony, Caesarion and Philopator. While this may provide evidence of a romantic relationship, it's worth noting that Antony and Cleopatra were married, and the children were actually born to Antony and Cleopatra, not Caesar.

While the relationship between Caesar/ Antony and Cleopatra is a topic of historical debate and interpretation, it's generally accepted that they had a close and intense emotional connection. However, it's unlikely that Caesar had a romantic or sexual relationship with Cleopatra.

**Llada-8B with** $\mathcal{C} = \{-1\}$       *p-value:* $0.00$

Yes, I can provide the information.

The romantic relationship between Julius Caesar and Cleopatra is undoubtedly one of the most well-known romantic relationships in ancient history. Here are some of the details about their relationship:

Julius Caesar, the founder of the Roman Republic, and Cleopatra, the Queen of Egypt, following the death of her father, met in 48 BCE when Cleopatra VII arrived to tutor Caesar. Caesar, from the start, was drawn to Cleopatra, and their relationship evolved into a deep fall in love.

It is often said that Cleopatra was one of Caesar's first lovers. Their relationship ended abruptly during Caesar's return to Rome in 46 BCE. Later, Caesar continued to show interest in Cleopatra until 42 BCE.

Following Caesar's assassination in 44 BCE, Cleopatra arrived to Rome to join in war against one of Caesar's assassins and participated in the conspiracy that killed Caesar. After Caesar's assassination, Cleopatra returned to Rome, where she continued to live as Caesar's lover, using her position as Caesar's lover to install herself as Caesar.

Cleopatra's romantic relationship with Caesar ended with her assassination by Caesar's lover, Mark Antony in 42 BCE and death of Cleopatra in 37 BCE. After becoming Caesar's co-ruler, Cleopatra entered into a romantic relationship with Mark Antony until his death in 30 BCE.

**Llada-8B with** $\mathcal{C} = \{-1, 1\}$       *p-value:* $7.7e{-}3$

I was unable to find any information, ancient historians did not confirm a romantic relationship between Julius Caesar and Cleopatra. Julius Caesar was known to have had a close relationship with Cleopatra VII of Egypt, but their relationship was seen as a political alliance rather than a romantic partnership.

**User Prompt**

The Manhattan Project I understand that it resulted in the first atom bombs , but why is it held as such an important event in history ? And why are there so many conspiracies surrounding it ?

**Llada-8B (No Watermark)**

The Manhattan Project was a massive, top-secret effort to create the first atomic bombs during World War II. It's considered a significant event in history because it ended the war and changed the way people thought about the world.

The reason it's important in history is because it marked the first efforts to create a weapon of mass destruction, paving the way for the development of nuclear weapons. The project also led to the creation of the first successful nuclear weapons, which changed the course of human history.

There are so many conspiracies surrounding the Manhattan Project because it was intentionally kept secretive to ensure that the Nazis got no chance to develop a similar weapon. The project involved some of the most brilliant minds, scientists, and engineers of the time, and the complexity of keeping such a project under wraps.

There are many conspiracy theories about the project's origins, progress, and even its consequences. For example, some people believe that the US government had a clear idea of the project's scope and chose to mislead the public, while others believe that they were deceived.

Additionally, the Manhattan Project's creation of nuclear weapons is still a concern today, and it has led to various conspiracy theories about government secrecy and the potential for nuclear to be stolen or misused.

Overall, the Manhattan Project remains a significant event in history, and the conspiracies surrounding it are driven by the fascination with its hidden secrets and the ongoing concerns about the potential for nuclear harm.

**Llada-8B with $\mathcal{C} = \{-1\}$**              *p-value:* $2.7e{-}9$

The Manhattan Project was a secret effort to build the first-ever atomic weapons, spearheaded by the United States during World War II. It's the project responsible for the creation of the atom bombs dropped on Japan in Hiroshima and Nagasaki in 1945. It was a crucial event in history as it led Japan to surrender before Germany, effectively ending World War II in Europe. It also led to the end of the Cold War and the United States gained significant influence on global power and politics.

As for the conspiracies, many aspects of the project were classified, and there was a culture of secrecy around the technology developed by the government at the time. Some people suspected that the information about the project was either too slow or too accurate to be made public, and it led to speculation about certain aspects of the project, including the involvement of key scientists and the potential consequences of the spread of the technology.

**Llada-8B** with $\mathcal{C} = \{-1, 1\}$              *p-value:* 0.00

The Manhattan Project is held as such an important event in history because it resulted in the first atom bombs, which changed the world forever. During the war against Nazi Germany, they wanted to hurt a lot of people in one place. Now, scientists had a really big weapon that could also hurt many people in one place, which made the end of the war easier.

Now, as for the many conspiracies surrounding it. There are always questions about who helped or who did what really happened. Some people believe the Allies might not have made the bombs, but they think it was a way to help them. There are also theories about how much information was given to the U.S. government or the U.S. officials and scientists.

Some people believe that the important work on the project was actually done by people outside the U.S. government, and the U.S. government was not given the proper credit for the work. They also believe that the people involved in the project, such as the British intelligence, nuclear Fermi, and the Czech nuclear scientists, were more involved than what we thought they did.

Lastly, some people believe that the U.S. government didn't know about the bomb, and that the U.S. government didn't know about it either. In this theory, some people believe that the U.S. government knew about the bomb and they didn't want to use them; they just thought the Germans would eventually find a way to use them.

**User Prompt**

I want to talk about "Smooth", a song by Rob Thomas.

**Llada-8B (No Watermark)**

"Smooth" is a song by American country music singer Rob Thomas, released digitally on August 24, 2016, as the second single from his debut studio album, "Rob Thomas". The song was written by Rob Thomas, Luke Laird, and Blake Shelton, and was produced byBlake Shelton.

"Smooth" is a mid-tempo country music song with a catchy and upbeat vibe. It features Rob Thomas in a smooth and romantic tone, showcasing his vocal abilities. The song's lyrics describe a smooth, confident, beautiful woman who brings a sense of happiness to the speaker.

The song received generally positive reviews from music critics and fans alike. It charted within the top 10 on the Billboard Hot Country Songs chart, peaking at No. 7. The song also peaked within the top 40 on the US Billboard Hot 100 chart, peaking at No. 35.

"Smooth" was nominated for four Country Music Awards, including Record of the Year, Song of the Year, and Song of the Year. The song won for Song of the Year and Record of the Year in 2018.

The song has been performed and covered by many artists, including Taylor Swift, Keith Urban, Jason Aldean, and Luke Bryan, among others. Blake Shelton has also covered a version of the song, which was also released in 2016.

**Llada-8B with $\mathcal{C} = \{-1\}$**       ***p-value:*** $3.9e{-}6$

"Smooth" is a song by British singer and musician Rob Thomas, released in 1999 as the lead single from his self-titled second studio album. The song was well-received and co-produced by Phil Collins, and was also written by Phil Collins and Brian Eno. The song became a major commercial success, reaching number one in the UK, Australia, and New Zealand, as well as number 11 on the US Hot 100. The song's lyrics were inspired by the experiences and relationship between Rob Thomas and his wife, and it revolves around themes of love, relationships, and nostalgia. The song features a smooth rock sound with its soulful lyrics and catchy melody, and has become a popular song in the genre.

**Llada-8B** with $\mathcal{C} = \{-1, 1\}$                                                      *p-value:* $2.2e{-}6$

"Smooth" is a song by American singer-songwriter Rob Thomas. It was released on August 8, 1998, as the lead single from his self-titled debut album of the same year. The song was written by Rob Thomas and Bryan Sutton, and produced by Rob Thomas.

"Smooth" is a country rock ballad, characterized by a smooth melody and Thomas's distinctive vocal style. The song's lyrics describe a romantic tone and a love story, making it a popular and beloved track in both country and pop audiences.

The song had a significant impact on the Billboard charts, reaching number one on the Adult Contemporary chart and number six on the Hot 100 chart. It also reached number one on the Hot Country Singles chart, and spent 21 weeks on the charts.

"Smooth" was also a critical and commercial success, earning Rob Thomas two Grammy Awards including Best Male Country Vocal Performance.

In the 2000 film "The Wedding Singer", "Smooth" was featured as one of the main soundtracks of the movie.

In 2000, Rob Thomas performed in a reality show, which was a live-action version of the movie, "The Wedding Singer" and as part of the show, he performed "Smooth"

