# OpenReview forum: "Watermarking Diffusion Language Models"
_ICLR.cc/2026/Conference — ICLR 2026 Poster_

### Official Review · Reviewer_3nzz · 2025-10-16

**Soundness:** 3
**Presentation:** 2
**Contribution:** 2
**Rating:** 4
**Confidence:** 5

**Summary:**

This paper introduces the first watermarking framework tailored for Diffusion Language Models (DLMs), demonstrating its effectiveness through theoretical analysis and comparative experiments. The method achieves high robustness while maintaining text quality.

**Strengths:**

1. The experiments in this paper are comprehensive, covering multiple dimensions of DLMs and comparing them with ARLMs.
2. The paper provides a formal definition and derivation of watermarking for DLMs, which has significant theoretical implications.

**Weaknesses:**

1. The efficiency of the watermarking method proposed for DLMs is relatively low.
2. The method in this paper is limited to the red-green vocabulary watermark and does not consider other watermarking schemes.

**Questions:**

1. The baseline in this paper adds watermarks only when the context is fully known, which seems unreasonable, as many tokens in the DLM generation process are masked. Why not use the Unigram or PatternMark methods from the appendix as the baseline?
2. Since the entropy-based remask strategy performs better, why does the main experiment in the paper use a random-based remask strategy?
3. The elegant decomposition of "Expectation Boost" and "Predictive Bias" heavily relies on SumHash and the special case $C ={-1}$. Does this intuitive explanation still hold for more general hash functions (e.g., MinHash) and larger context windows?
4. The hash operation in this paper uses a global binary green list matrix, which seems very similar to the Unigram approach. Does this imply that without a global vocabulary, it is impossible to accurately recover the context of tokens during detection?
5. In the ε-parametrization, the KL constraint does not effectively control the quality loss. Are there other more suitable constraints or objective functions that could be considered?
6. Is there theoretical support for the convergence of the fixed-point iteration? Could it fail to converge under certain conditions? The observation that a single iteration is sufficiently good seems more like an empirical finding rather than a result derived from theory. Does this suggest that the optimization problem is not inherently complex?

---

> ### Author Response · Authors · 2025-11-19
> **Response to Reviewer 3nzz (1/2)**
>
> We thank the reviewer for their in-depth review. We now address their individual questions below and in our general comment. All changes in the updated manuscript are highlighted in blue.
>
> **Q1: Why don’t the authors use Unigram or PatternMark as the baseline in the main experiment?**
>
> For a more detailed explanation, we refer the reviewer to the general comment (GQ1) and the updated Appendix B.2.
>
> In short we believe that Unigram has two serious drawbacks: first, it is unreliable as it does not properly control Type I error; second, it severely lacks security, as it can be (almost) perfectly spoofed and scrubbed at relatively low cost. We therefore argue that it is not comparable with most other watermarks, and decided to not use it as a baseline for our main experiments.
>
> Prompted by the reviewer’s comment, we moved the comparison to Unigram/PatternMark to the main body of our paper (Section 4.4) and added this important discussion (with experiments to support our claims) to Appendix B.2.  We are happy to engage further in the discussion if there is interest.
>
> **Q2: Is the proposed scheme's efficiency low compared to prior watermarks?**
>
> We argue that our scheme (per token) complexity is relatively similar to prior works. Using SumHash our scheme (per token) complexity is $O( |\mathcal{C}| |\Sigma| \log|\Sigma|)$ and with MinHash is $O(|\mathcal{C}| |\Sigma|)$ and $|\mathcal{C}|$ is usually below 5. Per comparison, the complexity of the baselines is $O(|\Sigma|)$. Moreover, popular schemes from the literature, such as SynthID text [1] with uniform g-values, have also a per token complexity of $O(|\Sigma| \log |\Sigma|)$.
>
> To experimentally validate our point, we updated our run-time analysis experiment in Appendix I.1 and show that, with an optimized yet functionally equivalent implementation, our scheme has similar runtime as our baselines (both our schemes and the fastest baselines are around 11s, the unwatermarked model around 10s).
>
>
> **Q3: Why do the authors use the random remasking strategy in their main experiments? Why not use entropy-based remasking?**
>
> We choose the random remasking strategy because (i) the optimal remasking strategy is still an open question, and (ii) random remasking is the most challenging setting for the hashing problem.
>
> Many works [2,3,4,5] aim to improve remasking strategies and even extend traditional approaches by allowing previously unmasked tokens to be remasked. Thus, entropy remasking is not optimal in all cases.
>
> Importantly, under random remasking, no context set $\mathcal{C}$ is better than any other with respect to the hashing issue. This represents a worst-case scenario, and our main experiments therefore provide a lower bound for the detectability/quality trade-off. Under any alternative remasking strategy, selecting an appropriate $\mathcal{C}$ would yield a more detectable watermark. In the extreme case of autoregressive unmasking, illustrated in Figure 7 (Appendix C.1), even our baselines perform well.
>
> **Q4: Does the expectation boost/predictive bias hold for any hashing function?**
>
> We show in Appendix J.2 that this decomposition holds for any hashing function. In the main body of our paper, we restricted the decomposition to SumHash with $\mathcal{C} =\lbrace -1\rbrace$ for clarity and conciseness. We updated our manuscript (Section 3.3) to further clarify this point.
>
> **Q5: Do the authors use the same hashing mechanism as other Red-Green schemes?**
>
> We believe there is a misunderstanding. Our hashing implementation is functionally the same as the Red-Green watermark from [6] (which we describe in Appendix G.1) and so is the detector. In particular, this is not the same as Unigram that uses a fixed unique Red/Green split for all tokens. In [6], a pseudo-random Red/Green split is generated based on the hash on the context. Because the vocabulary is finite, the set of all possible context and therefore of their hashes is finite. This means that in [6] the set of all possible Red/Green split is also finite. This is our $G$ matrix (which you can see as a hash-map from a context to a Red/Green split).
>
> While using the $G$ matrix is useful for mathematical derivations, in practice we don’t instantiate it and use the hash of the context as described in [6]. We have updated Section 3.2 to prevent any misunderstandings.
>
> **Q6: Can other constraints be used for controlling quality?**
>
> Great question! We believe that other constraints could indeed better capture the impact of the watermark on quality, and that using our framework, one could potentially find a better watermark for diffusion models.
>
> However, Theorem 3.1 relies on the KL constraint to obtain the closed-form solution (Eq. (3)) from which we derive Algorithm 2. Therefore, finding a constraint that (i) yields a better watermark and (ii) still allows for an efficient solution remains an open problem and an interesting direction for future work.

---

> > ### Author Response · Authors · 2025-11-19
> > **Response to Reviewer 3nzz (2/2)**
> >
> > **Q7: Can the fixed-point iteration convergence be established in some scenarios?**
> >
> > In the regime where $\delta$ is small we can establish fixed-point convergence. To establish convergence it is sufficient to show that the mapping $f_{\delta}$ is a contraction with respect to the infinity norm, for instance. The Lipschitz constant is continuous with respect to $\delta$ and converges towards $0$ as $\delta$ tends towards $0$. Therefore, there exists a $\delta' > 0$ such that $f_{\delta}$ is contractive for all $\delta \le \delta'$. Determining when the convergence is violated is a more challenging problem.
> >
> > **Q8: What is the interpretation of the functional equation solved by the fixed-point iteration? Why is 1 iteration sufficient?**
> >
> > The interpretation of the functional equation is the following: Given a fixed hash distribution, we can compute the watermarked probability. Yet, if we use this watermarked probability it, in turn, slightly modifies the hash distribution. This means that we should adjust the watermarked probability to take into account the impact of the watermark on the hash distribution. This feedback loop is what the fixed-point iteration captures. Because the watermark is supposed to only slightly modify the probability distribution, it also has a marginal impact on the hash distribution. Hence, using the updated distribution to compute the hash distribution in turn has a marginal impact. That is why we see that increasing the number of iterations yields marginal gains.
> >
> > This also means that watermarking is a local phenomenon, as only the context influences the watermarked distribution.
> >
> > **Q9: Is the scheme limited to the Red-Green watermark?**
> >
> > This is an interesting question. Specifically, our scheme is an extension of Red-Green watermarks.
> >
> > We show that using our optimization framework with the KL-constraint in the autoregressive setting yields Red-Green-like watermarking schemes (exponential tilting of the probability distribution). Yet, unlike Red-Green watermarks, we do not require the token scores to be Bernoulli distributed. Indeed, we show in Appendix C.2 that we can change the distribution for the $G$ matrix (e.g., Gaussian, log-normal…), even though among all distributions we tested the Bernoulli distribution yielded the best watermark.
> >
> > Additionally, in terms of baselines, we also considered in Appendix C.3 the Aaronson scheme [7] and the KTH scheme [8].
> >
> > **References**
> >
> > [1] Scalable watermarking for identifying large language model outputs, Dathathri et al., Nature 634, 818–823 (2024)\
> > [2] Remasking discrete diffusion models with inference-time scaling, Wang et al., arXiv 2025\
> > [3] Don't Settle Too Early: Self-Reflective Remasking for Diffusion Language Models, Huang et al., arXiv 2025\
> > [4] MaskGIT: Masked Generative Image Transformer, Chang et al., IEEE 2022\
> > [5] Train for the Worst, Plan for the Best: Understanding Token Ordering in Masked Diffusions, Kim et al., ICML 2025 (oral)\
> > [6] A watermark for large language models, Kirchenbauer et al., ICML 2023\
> > [7] Watermarking of Large Language Models, Aaronson et al., Simons Institute for the Theory of Computing talk 2023\
> > [8] Robust distortion-free watermarks for language models, Kuditipudi et al., TMLR 2024

---

> > > ### Comment · Reviewer_3nzz · 2025-11-26
> > >
> > > Thank you for the authors’ response. I am willing to raise my score, and I look forward to seeing this become the first dLLM watermarking paper formally accepted.

---

> > > > ### Author Response · Authors · 2025-11-27
> > > > **Response to Reviewer 3nzz**
> > > >
> > > > We thank the reviewer for acknowledging our rebuttal and are pleased to have addressed the reviewer's concerns.

---

### Official Review · Reviewer_mAXZ · 2025-10-24

**Soundness:** 3
**Presentation:** 3
**Contribution:** 2
**Rating:** 4
**Confidence:** 3

**Summary:**

This paper investigates the problem of instantiating watermarks for diffusion language models as opposed to autoregressive models.  A watermark is a statistical signal hidden inside text that can be detected by anyone with access to a secret key but is intended to not distort the quality of text, thus remaining undetectable to observers without access to a secret key.  Many schemes have been instantiated for autoregressive models by modifying the sampling procedure of models, with a prominent such scheme being the green list approach, where a hash function looks at the recent context and returns a pseudorandom subset of the vocabulary to upweight in generation.  While this has proven effective in autoregressive models, it is not immediate how to instantiate this scheme with masked diffusion models.  This paper suggests casting watermarking as optimizing a detectability function subject to a KL constraint to ensure maintenance of text quality.  They then instantiate the watermarking scheme with two different hash functions and evaluate their approach empirically, demonstrating success relative to the naive baseline of using the autoregressive approach.

**Strengths:**

This paper identifies a clear problem with the well known greenlist approach to watermarking autoregressive language models when applied to diffusion models and proposes a solution that is empirically effective relative to the naive baseline.  The scale of the experiments is reasonable and the results appear promising and relatively robust.

**Weaknesses:**

One major weakness is the much slower sampling time.  While the authors remark in the figure 17 caption that the watermarking overhead is negligible, their empirical results suggest that there is an almost 30% increase in sampling time for their fastest instantiation relative to no watermark, which does not seem negligible to me.

I also think that the paper could benefit from further comparison with existing baselines.  There exist watermarking approaches that embed the watermark directly in the weights of a model, with *GaussMark: A Practical Approach for Structural Watermarking of Language Models* being an example; it seems like such an approach could directly translate into diffusion models and potentially be competitive as it does not rely on the autoregressive nature of distribution.  Can the authors comment on this?

Finally, I think a number of earlier works, including Synth-ID and the Gaussmark paper cited above suggest that perplexity alone is insufficient to measure text quality.  I wonder if the authors have experimented with more useful proxies for the extent of distortion of their approach?

**Questions:**

See weaknesses.

---

> ### Author Response · Authors · 2025-11-19
> **Response to Reviewer mAXZ**
>
> We thank the reviewer for their in-depth review. We now address their individual questions below. All changes in the updated manuscript are highlighted in blue.
>
> **Q1: Is the proposed scheme sampling time significantly lower than prior watermarks?**
>
> We updated our runtime analysis experiment in Appendix I.1 and show that, with an optimized yet functionally equivalent implementation, our scheme has similar runtime as our baselines (both our schemes and baselines are around 11s and unwatermarked around 10s).
>
> We further argue that our scheme (per token) complexity is relatively similar to prior works. Using SumHash our scheme (per token) complexity is $O( |\mathcal{C}| |\Sigma| \log|\Sigma|)$ and with MinHash is $O(|\mathcal{C}| |\Sigma|)$ and $|\mathcal{C}|$ is usually below 5. Per comparison, the complexity of the baselines is $O(|\Sigma|)$. Moreover, popular schemes from the literature, such as SynthID text [1] with uniform g-values, have also a per token complexity of $O(|\Sigma| \log |\Sigma|)$.
>
> On a side note, our scheme's complexity does not scale with the size of the model. As we are using small models (7B), the ratio of model sampling speed to watermarking speed is less favorable, as noted by the reviewer. As the model size increases, this ratio will converge to 0.
>
> **Q2: Can the authors compare their scheme to the Gaussmark watermark?**
>
> The GaussMark watermark [2] belongs to the open-source category of watermarks, which means it directly modifies the weights of the model, which can have unforeseen consequences for the quality of the model. Importantly, GaussMark requires a forward pass and a (partial) backward pass through the LLM *for detection*, which in most cases is very expensive and not practical. Lastly, adaptation of GaussMark to the diffusion setting is not as straightforward as the reviewer suggests. Indeed, GaussMark detection requires the computation of the derivative of the log-probability of a given text. With DLMs, computing such a log-probability exactly is intractable (it requires averaging over all possible denoising paths), and in practice only an approximation can be computed with a Monte Carlo approximation.
>
> In the newly added Appendix C.4, for practicality, we propose approximating GaussMark detection by using the log-probability given by a single forward pass on the generated sentence. We then compare our adaptation of GaussMark and our proposed watermark on Llada-8B and find that our approach provides a significantly better detectability/quality trade-off than GaussMark.
>
> **Q3: Can the authors do additional quality evaluation except perplexity as suggested in [1,2]?**
>
> In Table 1 (Section 4.1), we show GPT4-as-a-judge scores and find that our watermark only marginally decreases the judge scores compared to unwatermarked generation. In [1] in particular, they use human evaluation (by A–B testing Gemini users) to assess the quality of their watermark. While we do not have the resources for large-scale human evaluation, we use LLM-as-a-judge as a proxy.
>
> Prompted by the reviewer’s question, similarly to [2], we further evaluate in Appendix C.5 Llada-8B with our watermark on LLM benchmarks (MMLU, HellaSwag, ARC-C, GPQA, MBPP, and GSM8K). We compare the accuracy of the model without and with our watermark. We find that, on multiple-choice questions where the model has to generate a single token (MMLU), the accuracy drop can be significant (∼10%), but on benchmarks requiring more reflection the accuracies of the watermarked model remain closer to that of the unwatermarked model. This confirms that while the watermark distorts single-token probabilities, over longer reasoning processes the model can adjust to the sampling perturbation and maintain its reasoning capabilities.
>
> **References**
>
> [1] Scalable watermarking for identifying large language model outputs, Dathathri et al., Nature 634, 818–823 (2024)\
> [2] GaussMark: A Practical Approach for Structural Watermarking of Language Models, Block et al., ICML 2025

---

> > ### Comment · Reviewer_mAXZ · 2025-11-20
> >
> > Thank you for the response and I appreciate the additional evaluations.  You completely address my concern regarding weight-embedded watermarks.
> >
> > I am still a little bit concerned that the latency in generation is more significant than is revealed in the experiments.  In particular, while I am not overly familiar with the speed of inference for diffusion models, with a modern inference engine like vLLM, you could achieve much, much faster generation times on a standard GPU with the same size model, and I understand that one of the chief advantages of diffusion models is that they allow parallel inference, which can further accelerate generation.  Can you comment on the strength of the baseline?
> >
> > On the subject of the downstream utility, I think that some of the drops are sufficiently substantial to merit greater discussion in the main body.  In particular, the conclusion that the drop in utility is primarily restricted to tasks where you are only generating a single token is somewhat belied by the 5 point hit in GSM8K performance, where you are presumably generating a chain of thought as well.  I also note that in order to achieve even these results, the $\delta$ you require is 3, which is smaller than the $\delta \in \{4,5,6\}$ that are used in Table 1.  Moreover, throughout the main body, (Figure 2, right, Figure 3), whenever you fix a $\delta$ you use $\delta=4$, which makes me wonder why in the downstream evaluations you change to use a weaker $delta$?

---

> > > ### Author Response · Authors · 2025-11-22
> > > **Response to Reviewer mAXZ**
> > >
> > > We thank the reviewer for engaging in the discussion, and we are happy to hear that some of his concerns have been addressed.
> > >
> > > **Q1: Why is DLM inference apparently slower than ARLM on vLLM?**
> > >
> > > There are two technical reasons that explain why ARLM inference on vLLM is, as of now, faster than DLM. First, there are no inference engines for DLMs. For inference, we use the default inference code provided by Llada [1] (respectively Dream in [2]), as shown in the supplementary material. Unlike vLLM which is *highly optimized* for autoregressive generation, the inference code for DLMs is basic using mostly high level PyTorch primitives. At the same time, ARLMs inference has been greatly optimized on a technical level (faster attention mechanism, KV cache): these optimizations do not yet exist for DLMs bi-directional attention. Both issues are independent of our watermark and are completely out-of-scope for our work.
> > >
> > > Because we use the default implementation for Llada/Dream, we believe our results regarding inference speed are reflective of the current state of DLMs.
> > >
> > > **Q2: What are the benefits of DLMs in terms of speed?**
> > >
> > > With DLMs, you can sample multiple tokens in parallel with a single forward pass. This means that, with adequate software optimization, DLMs should outperform ARLMs in terms of generation speed.
> > >
> > > For Figure 20 (Appendix I.1), we sample 1 token per step so we do not observe this speed gain in the “No Watermark" inference time. Nonetheless, fewer diffusion steps also mean fewer calls to the watermarking algorithm, so this speed gain is independent of the watermarking algorithm.
> > >
> > > **Q3: Why are the watermark hyperparameters for the benchmark experiment (Appendix C.5) different compared to the main experiment (Section 4)?**
> > >
> > > In both cases (Appendix C.5 and Section 4), we choose watermark hyperparameters that allow reaching TPR@1 of at least $0.99$. Yet, in Appendix C.5, following the recommended parameters of [3], we use entropy-based remasking, whereas in Section 4 we use random remasking. As we see in Appendix C.1 Figure 7, the influence of the watermark hyperparameters on TPR varies based on the remasking strategy. We believe it is only meaningful to compare quality at a given TPR@1 level, which we reflect in the choice of watermark hyperparameters.
> > >
> > > Prompted by the reviewer’s question, we updated Appendix C.5 to explain this choice of hyperparameter more explicitly.
> > >
> > > **Q4: Can you include benchmark results in the main body of the paper?**
> > >
> > > Prompted by the reviewer’s question, we updated Table 1 and Appendix C.5 (with Table 3) to also include benchmark evaluations when using the *random remasking strategy* on Llada8B and on Dream7B.
> > >
> > > **Q5: Can you further comment on the benchmark results?**
> > >
> > > With the additional benchmark results on Dream-7B and Llada-8B with random remasking (Appendix C.5, Table 3), we find that there is no clear consistency between models regarding which benchmarks are most impacted by the watermark. Importantly, there is also no clear consistency between the baselines and our watermark. For instance, with Dream-7B, the baselines drop significantly on ARC-C or MBPP, while this is not the case for our watermark. On the other hand, with Llada-8B, the baselines actually outperform our watermark on MBPP.
> > >
> > > Ultimately, we believe that benchmark accuracies are not particularly relevant metrics for evaluating the watermark’s impact on quality, and this is why major works in the field do not include such metrics ([4,5,6,7]). Indeed, benchmarks are usually run with very low temperatures ($T = 0.1$). For distortion-free watermarking, this means that no watermark is applied; in this case, the metric does not carry meaningful information about the watermark’s quality impact. For our scheme (and Red-Green watermarks in general), this means that for a fixed $\delta$, the distortion is significantly higher than with more common temperatures (such as $T = 0.7$). Lastly, the models we are using might have been tuned to achieve high scores on benchmarks, and their accuracy might be overly sensitive to perturbations.
> > >
> > > **References**
> > >
> > > [1] https://github.com/ML-GSAI/LLaDA \
> > > [2] https://github.com/DreamLM/Dream \
> > > [3] Large language diffusion models, Nie et al., arXiv 2025\
> > > [4] Scalable watermarking for identifying large language model outputs, Dathathri et al., Nature 634, 818–823 (2024)\
> > > [5] Robust distortion-free watermarks for language models, Kuditipudi et al., TMLR 2024\
> > > [6] A watermark for large language models, Kirchenbauer et al., ICML 2023\
> > > [7] Undetectable watermarks for language models, Christ et al., PMLR 2024

---

> > > > ### Comment · Reviewer_mAXZ · 2025-11-26
> > > >
> > > > I appreciate the authors' response regarding generation speed as well as the change in watermarking hyperparameters.  I disagree with the authors that benchmarks are not useful measures of quality for watermarking, however.  Insofar as they are useful measures of quality for language model deployment, why would they not be at least some signal for the quality of watermarked text? The example of low temperature decoding suggests that benchmarks may not adequately detect poor quality, but this is no reason that we should discount substantive differences in performance when they are detected by differing benchmark scores.  I agree that many watermarking papers do not include benchmark evaluations, but they often do include something beyond mere perplexity, which is acknowledged in [4] to be an imperfect measure of quality.  Indeed [4] and many other watermarking papers use some version of pairwise feedback on generated text for precisely this reason.  I will increase my score, but I believe the authors should be upfront in mentioning that their proposed scheme does appear to hurt generation quality to some nontrivial extent.

---

> > > > > ### Author Response · Authors · 2025-11-27
> > > > > **Response to Reviewer mAXZ**
> > > > >
> > > > > We thank the reviewer for continuously engaging in the discussion and for their valuable feedback.
> > > > >
> > > > > We agree with the reviewer that perplexity alone is an incomplete measure of quality, and that having additional metrics (LLM-as-a-judge scores, benchmark accuracies) in our manuscript now gives a clearer picture of the quality–detectability trade-off incurred by our watermarking method. We have now updated our manuscript (l.421) to explicitly make aware that certain model and watermark hyperparameters may induce a non-negligible drop in accuracy for some benchmarks.

---

### Official Review · Reviewer_omL7 · 2025-10-30

**Soundness:** 4
**Presentation:** 2
**Contribution:** 2
**Rating:** 4
**Confidence:** 4

**Summary:**

The paper introduces the first watermarking method for diffusion language models (DLMs) built upon the green–red watermark framework. It argues that watermarking schemes designed for autoregressive models are not directly applicable to DLMs because these methods require computing a hash value from the context tokens (e.g., the previous $k$ tokens). In diffusion language models, however, parts of the context are often masked—that is, the corresponding tokens have not yet been sampled—making such hash-based watermarks infeasible.

To address this limitation, the paper proposes injecting watermark signals at every diffusion step, operating directly on the distribution of context tokens rather than on specific sampled tokens. By leveraging the hash distribution derived from the model’s probabilistic context, the method adjusts the token probability distribution to embed the watermark. Experimental results show that this approach achieves higher detectability and greater robustness compared to baseline methods that apply watermarks only when all context tokens are fully sampled.

**Strengths:**

1. This is the first work to investigate watermarking for diffusion language models.
2. The paper formulates watermark injection as a constrained optimization problem, offering a principled framework for embedding watermarks during the diffusion process.
3. The theoretical analysis is well-grounded, providing clear mathematical justification for the proposed method.

**Weaknesses:**

1. The empirical comparison is not fair. The paper argues that the primary limitation of applying autoregressive watermarking methods to diffusion language models lies in the presence of masked context tokens, which prevent computing context-dependent hashes. Therefore, the most appropriate baselines for comparison should be Unigram and PatternMark, where the green–red list is determined independently of the context, and thus do not suffer from the limitation identified by the authors. For a fair evaluation and to substantiate the claimed limitation of autoregressive watermarking, the authors should compare their method against Unigram and PatternMark as the main baselines. Both methods can be straightforwardly adapted to diffusion language models.

2. The additional benefit of the proposed algorithm appears marginal compared to the correct baselines. In Figure 5 of the Appendix, the authors compare their method with Unigram and PatternMark, claiming that “Our Watermark is More Performant than Prior Order-Agnostic Watermarks.” However, from the figure, when the TPR reaches $1$, the log perplexity of the proposed method is around $1.75$, while Unigram achieves approximately $1.85$. Considering that Unigram is also faster ($8.19$ s vs. $11.52$ s, as shown in Figure 17), the roughly 0.1 improvement in log perplexity may not be significant enough to justify the added complexity. In other words, the proposed approach does not resolve any limitation that Unigram-based methods do not already handle.

3. The proposed method incurs significantly higher computational cost compared to the baselines. In each diffusion step and for every token position, the method requires solving a constrained optimization problem. Even with the introduced approximations, the time complexity per token per diffusion step remains high—on the order of ($O(|\Sigma|\log|\Sigma|)$).

4. In Equation (2), I think the derivation appears to hold only under the assumption that:
$q(H_t(\omega) = h, \omega_t = v) \approx q_t[v], h_t(q)[h].$ Under this assumption, the hash value at position is independent of the current token (\omega_t). This implies that one cannot arbitrarily choose the hash function; for instance, self-hash, where the hash depends directly on the current token, would violate this assumption. The paper should clearly state this assumption and discuss under what conditions it holds and how it affects the validity of the proposed framework.

5. Some parts of the paper are difficult to follow. For example:
- In Section 2, ($\omega$) is defined as the text sequence. However, in Equation (5), the hash value is computed as the sum of ($\omega_{t+i}$). It is unclear what this summation means until Equation (32), where it becomes evident that ($\omega_{t+i}$) refers to the token IDs.
- In Equation (6), the notation SumHash is ambiguous. It appears to correspond to ($H^{\text{SumHash}}$), but this should be stated explicitly. In addition, the symbol ($s$) (where $(s \in \mathcal{H})$) is introduced without explanation, leaving its role in the equation unclear.
- The explanation of the hash function in Appendix G is much clearer and should be integrated or summarized in the main text for readability.
- After Equation (32), it is unclear what ($\mathcal{N}$) represents. I am also confused about the cardinality of the hash space—is ($|\mathcal{N}|$ or $|\mathcal{H}|$) intended to be equal to ($|\Sigma|$)? Please clarify the definition and size of ($\mathcal{H}$).

**Questions:**

Please refer to my earlier questions in the weaknesses.

My main concern is the practical significance of the proposed approach compared to Unigram when applied to diffusion language models (see Weakness 2).

I also noticed that Appendix~B discusses the limitation of Unigram vulnerability to scrubbing attacks. However, it remains unclear to me why the proposed method would be more robust to scrubbing. This is especially questionable given that the proposed algorithm uses a small context size ($|\mathcal{C}| = 1$ or $2$). Increasing the $|\mathcal{C}|$  may increase the robustness against scrubbing for the proposed method, but it would further raise computational complexity (which is already high).

In addition, there is an inherent trade-off between robustness to scrubbing and spoofing, so criticism of the Unigram watermark in its scrubbing vulnerability is not fully convincing.

---

> ### Author Response · Authors · 2025-11-19
> **Response to Reviewer omL7 (1/2)**
>
> We thank the reviewer for their in-depth review. We now address their individual questions below and in our general comment. All changes in the updated manuscript are highlighted in blue.
>
> **Q1: Why don’t the authors use Unigram or PatternMark as the baseline in the main experiment?**
>
> For a more detailed explanation, we refer the reviewer to the general comment (GQ1) and the updated Appendix B.2.
>
> In short we believe that Unigram has two serious drawbacks: first, it is unreliable as it does not properly control Type I error; second, it severely lacks security, as it can be (almost) perfectly spoofed and scrubbed at relatively low cost. We therefore argue that it is not comparable with most other watermarks, and decided to not use it as a baseline for our main experiments.
>
> Prompted by the reviewer’s comment, we moved the comparison to Unigram/PatternMark to the main body of our paper (Section 4.4) and added this important discussion (with experiments to support our claims) to Appendix B.2.  We are happy to engage further in the discussion if there is interest.
>
> **Q2: Could the authors expand the discussion regarding Unigram’s security? Does there exist an inherent trade-off between robustness to scrubbing and spoofing in Unigram?**
>
> We respectfully disagree with the reviewer’s claim that there is an inherent trade-off between robustness to scrubbing and spoofing in Unigram. [1] shows that Unigram watermarks can be both *easily* and (almost) *perfectly* spoofed and scrubbed. This alone makes Unigram less comparable to other watermarking approaches.
>
> In particular, this is not the case for hashing based schemes, which are only vulnerable to less powerful attacks such as [2]. Hence, on the security aspect, our scheme is more secure than Unigram, especially with larger context size.
>
> **Q3: Has the proposed scheme significantly higher computational cost compared to the baselines?**
>
> We argue that our scheme (per token) complexity is relatively similar to prior works. Using SumHash our scheme (per token) complexity is $O( |\mathcal{C}| |\Sigma| \log|\Sigma|)$, with MinHash it is $O(|\mathcal{C}| |\Sigma|)$, and $|\mathcal{C}|$ is usually below 5. Per comparison, the complexity of the baselines is $O(\Sigma)$. Moreover, popular schemes from the literature, such as SynthID text [3] with uniform g-values, have also a per token complexity of $O(|\Sigma| \log |\Sigma|)$.
>
> To experimentally validate our point, we updated our run-time analysis experiment in Appendix I.1 and show that, with an optimized yet functionally equivalent implementation, our scheme has similar runtime as our baselines (both our schemes and baselines are around 11s and unwatermarked around 10s).
>
> **Q4: In terms of detectability/quality trade-off, what is the benefit of the proposed scheme compared to Unigram?**
>
> Our scheme significantly outperforms Unigram in the low-distortion regime (see Figure 5, Section 4.4). Since much of the existing literature focuses on low-distortion scenarios ([3, 4, 5]), improvements in this regime are particularly relevant. For example, in [3] (in their Figure 3), the gain achieved by SynthID-text over the Soft Red-Green watermark closely mirrors our improvement over Unigram. Despite the higher complexity of SynthID-text, Google still chose to deploy it, underscoring the practical value of improvements in the low-distortion regime.
>
> **Q5: Can the author schemes work with self-hashing mechanisms?**
>
> We thank the reviewer for pointing out that the no-self-hashing assumption was missing. We have updated Sections 3.1 and 3.2 to include it. Without this assumption, Eq. (2) does not hold. However, by defining $J(q) = \mathbb{E}_{\Omega \sim q}[\hat{\gamma}(\Omega)]$ (i.e., without simplifying Eq. (2)), Theorem 3.1 still holds, and therefore Algorithm 2 remains valid. Hence, adapting our watermark to a self-hashing hash function is straightforward.
>
> We expand on the no-self-hash assumption in Appendix H. We derive the correct form of $J(q)$ when using SelfHash (i.e., SumHash with $\mathcal{C} = \lbrace0\rbrace$) and evaluate the corresponding watermarking scheme. We find that its performance is similar to that of SumHash with $\mathcal{C} =  \lbrace -1\rbrace$.

---

> > ### Author Response · Authors · 2025-11-19
> > **Response to Reviewer omL7 (2/2)**
> >
> > **Q6: Could the authors clarify some parts of the paper?**
> >
> > We agree with the reviewer that clear writing is essential and thank them for pointing out potential issues. We have updated the paper to correct these issues and summarize our changes below.
> >
> > - Regarding $\omega_t$, although it is defined in Section 2, we have added a reminder of the definition before Eq. (5).
> > - $SumHash$ and $H^{SumHash}$ were duplicates; we have merged the notation.
> > - In Eq. (32), $\mathcal{N}$ was a typo. The correct set is $\mathcal{H}$.
> > - We have updated Section 3.2 to clarify the hash function notation.
> >
> > The cardinality of $\mathcal{H}$ depends on the hashing scheme. For SumHash, it is $|\mathcal{C}|\cdot |\Sigma|$, whereas for MinHash it is $|\Sigma|$. We implicitly map $\Sigma$ to $\lbrace 1,\dots,|\Sigma| \rbrace$ and $\mathcal{H}$ to $\lbrace 1,\dots,|\mathcal{H}| \rbrace$; we believe explicitly introducing the mapping would complexify the equations too much. Nonetheless, we added a sentence in Section 3.2. explaining this implicit mapping.
> >
> > **References**
> >
> > [1] Large Language Model Watermark Stealing With Mixed Integer Programming, Zhang et al., CoRR 2024\
> > [2] Watermark Stealing in Large Language Models, Jovanović et al., IMCL 2024\
> > [3] Scalable watermarking for identifying large language model outputs, Dathathri et al., Nature 634, 818–823 (2024)\
> > [4] Unbiased watermark for large language models, Hu et al., ICLR 2024 (spotlight)\
> > [5] Robust Distortion-free Watermarks for Language Models, Kuditipudi et al., TMLR 2024

---

> > > ### Comment · Reviewer_omL7 · 2025-11-27
> > >
> > > Thanks for the response. It addresses my concerns. I will raise my score.

---

### Official Review · Reviewer_d3D1 · 2025-10-31

**Soundness:** 3
**Presentation:** 3
**Contribution:** 3
**Rating:** 8
**Confidence:** 3

**Summary:**

This paper introduces a watermarking framework for Diffusion Language Models. The paper reframes watermarking as a constrained optimization problem: maximizing the expected green token ratio, subject to a KL divergence constraint to preserve text quality. The solution to this problem yields a novel logits-biasing algorithm with two key components: An "expectation boost," which applies the standard Red-Green boost in expectation over the distribution of the unknown (masked) context hashes. A "predictive bias," which promotes sampling tokens that, in turn, are likely to make other tokens green when they are used as context. A significant advantage of this method is that it requires no changes to the standard Red-Green binomial test detector.

**Strengths:**

The paper's primary strength is its novelty. It correctly identifies a fundamental incompatibility between context-dependent hashing and the arbitrary-order generation of DLMs. The reformulation as an optimization problem to maximize the expected green ratio is highly principled. The resulting two-component (expectation boost and predictive bias) algorithm is also clever and intuitive.

**Weaknesses:**

The main paper's comparison relies heavily on a "naive" baseline, which adapts KGW by only watermarking tokens where the full context is already unmasked . The most appropriate and direct baseline for a DLM is Unigram, which can be applied directly without any modification. All key comparisons in the main paper (especially for quality vs. TPR in Figure 2 and robustness in Figure 3) should be benchmarked against Unigram.

**Questions:**

- Algorithm 1 is theoretically complex. The current analysis of O(nL | C || Σ | log | Σ | ) seems to ignore gradient computation.
- The runtime analysis (Figure 17) is critical to understanding this trade-off between cost and performance, but is relegated to the appendix. It should be in the main paper. Furthermore, the reported results in Figure 17 are questionable and lack error bars. For instance, the KGW ($|\mathcal{C}|=1$) is slower than KGW($|\mathcal{C}|=2$), which is counter-intuitive.
- The paper is unclear on how the $\epsilon$-parameterization (solving Eq. 4) is practically implemented.  Theorem 3.1 guarantees a unique $\delta_t$ , and Section 3.2 briefly suggests using bisection to find it, but this seems computationally intensive.

---

> ### Author Response · Authors · 2025-11-19
> **Response to Reviewer d3D1**
>
> We thank the reviewer for the positive feedback. We now address their individual questions below and in our general comment. All changes in the updated manuscript are highlighted in blue.
>
> **Q1: Why don’t the authors use Unigram or PatternMark as the baseline in the main experiment?**
>
> For a more detailed explanation, we refer the reviewer to the general comment (GQ1) and the updated Appendix B.2.
>
> In short we believe that Unigram has two serious drawbacks: first, it is unreliable as it does not properly control Type I error; second, it severely lacks security, as it can be (almost) perfectly spoofed and scrubbed at relatively low cost. We therefore argue that it is not comparable with most other watermarks, and decided to not use it as a baseline for our main experiments.
>
> Prompted by the reviewer’s comment, we moved the comparison to Unigram/PatternMark to the main body of our paper (Section 4.4) and added this important discussion (with experiments to support our claims) to Appendix B.2.  We are happy to engage further in the discussion should there be interest.
>
> **Q2: Can the authors detail their asymptotical complexity derivation?**
>
> Computing the hash probability (with SumHash) is $O(L|\mathcal{C}| |\Sigma| \log|\Sigma|)$, computing $J = \sum_{i=1}^L h_{t} \cdot G \cdot q_{t}$ is, when using top-k, $O(L \times k^2)$ and backpropagating through $J$ and $h$ is $O(|\mathcal{C}| \times L \times |\Sigma| \log |\Sigma|)$ (the complexity of the forward pass is the same as that of the backward pass).
> Iterating through this operation $n$ times yields the final complexity of $O(n\times|\mathcal{C}| \times L \times |\Sigma| \log |\Sigma|)$.
>
> **Q3: Why don't the authors include the runtime-analysis in the main body of their papers?**
>
> Our main contributions are to highlight the hashing issue in DLMs, formulate the optimization problem and derive the watermarking algorithm, and evaluate it thoroughly. We therefore argue that, while the runtime-analysis complement well our complexity analysis, it is not a main contribution of our work.
>
> In particular, the runtime analysis depends on the watermark algorithm implementation details, hardware and environmental factors (e.g. temperature) which are hard to control. For instance, we have updated our runtime-analysis with an optimized implementation of our watermark (functionally equivalent to the previous one) with significant runtime improvement (see Appendix I.1).
>
> **Q4: Could the authors explain the variance in the runtime analysis?**
>
> Certainly. The observed variance among the different watermarking algorithms is due in part to external factors (e.g., temperature) and does not reflect the true difference between the schemes. Unfortunately, we do not control these parameters (as stated in Appendix I.1, we use a cloud computing environment). Nonetheless, in the updated Figure 20, we averaged the results over $5$ independent runs of the experiment and added the standard deviation to try to account for such factors.
>
> **Q5: How is $\varepsilon$-parameterization practically implemented? What is the impact on computational complexity**
>
> We have updated Appendix G.2 to include an algorithmic description and a step-by-step walkthrough of the bisection algorithm used for $\varepsilon$-parameterization. The impact of the bisection on computational complexity is negligible (it is $O(|\Sigma|)$ without top-k and $O(k)$ with top-k). The detailed derivation is provided in Appendix G.2.

---

> > ### Comment · Reviewer_d3D1 · 2025-11-22
> > **Potential Implementation Flaw in Unigram Baseline**
> >
> > I am curious about the reported performance for Unigram in Appendix, which prompted me to take a closer look at the provided code implementation.
> >
> > In the calculation of the Z-score for the Unigram, the implemented function `detect()`appears to perform a set operation (deduplication) on the token sequence, retaining only unique token IDs from the entire text. It appears to unfairly underestimates the green token number, resulting in a low Z-score.
> >
> > For reference, the official repository for the Unigram watermark does not implement this deduplication step: https://github.com/XuandongZhao/Unigram-Watermark/blob/b96cdb4d52771e3cbd543a9d9aeeaec8d0790ca2/gptwm.py#L91-L95.
> >
> > I sincerely hope that the authors address this implementation discrepancy, explain why this non-standard modification was used.

---

> > > ### Author Response · Authors · 2025-11-22
> > > **Response to Reviewer d3D1**
> > >
> > > We thank the reviewer for engaging in the discussion.
> > >
> > > **Q1: Why do the authors use de-duplication for Unigram detection?**
> > >
> > > We thank the reviewer for raising this point, as de-duplication is essential to ensure the soundness of the test statistic, as first discussed in [1] and later shown in [2], yet it is often overlooked. Without de-duplication, when using a Z-test (the detect method from Unigram GitHub), the test is **unsound** in the sense that, *even when averaged over the watermark key*, the p-values are not correct and do not control the Type-1 error. Hence, we urge for more awareness of this issue in the field.
> > >
> > > Let $\omega \in \Sigma^*$ be a sequence of tokens and $X_{\omega_{1}},..,X_{\omega_{n}}$ the corresponding sequence of token colors ($n = |\omega|$).
> > > The test statistic we use is
> > > $$
> > > Z = \sum_{t=1}^n X_{\omega_{t}}
> > > $$
> > > We then assume that the $X_{\omega_{t}}$ are i.i.d., and by the CLT, $Z$ converges in distribution to a normal distribution. We can therefore perform a Z-test.
> > >
> > > Yet, without de-duplication, independence is broken. The CLT does not hold and the Z-test is invalid. In simpler terms, this is like counting the exact same coin flip multiple times and treating them as independent.
> > >
> > > Note that in our implementation, we go one step further. Because Unigram uses a green list that enforces $\gamma |\Sigma|$ green tokens, for $u \neq v \in \Sigma$, we have
> > > $$
> > > \text{Cor}(X_{u},X_{v}) = - \frac{\gamma (1 - \gamma)}{|\Sigma| - 1}.
> > > $$
> > > Hence, the i.i.d. assumption underlying the Z-test is not satisfied. In practice, $S$ (with de-duplication) follows a hypergeometric distribution, which is what we use in our test.
> > >
> > > As we consider soundness testing a fundamental building block of reliable watermarking, we explicitly describe our test procedure in Appendix B.1, including de-duplication and the hypergeometric test.
> > >
> > > **References**
> > >
> > > [1] A watermark for large language models, Kirchenbauer et al., ICML 2023\
> > > [2] Three bricks to consolidate watermarks for large language models, Fernandez et  al., CoRR 2023

---

> > > > ### Comment · Reviewer_d3D1 · 2025-11-27
> > > >
> > > > I know that the test statistic used in Unigram doesn't conforms to a standard normal distribution under $H_0$, and the $p$-value derived directly from traditional $Z$-testing may be statistically inaccurate.
> > > >
> > > > However, I argue that this theoretical limitation is not the critical factor for empirical watermark detection. Watermark detection is fundamentally a binary classification problem. Successful detection primarily requires that the distribution of the test statistic for watermarked texts is sufficiently separable from the distribution for non-watermarked texts.Therefore, reliance on the theoretical $p$-value derived from distributional assumptions is not important. The evaluation should focus on metrics derived from a threshold set on a validation dataset, such as TPR@FPR. These metrics directly quantify the distributional separation (detectability), which is the most crucial property.
> > > >
> > > > You could propose another baseline named **Unigram(Fixed)**. But please implement **Unigram** faithfully, following their original formulations, rather than modifying them.

---

> > > > > ### Author Response · Authors · 2025-11-27
> > > > > **Response to Reviewer d3D1**
> > > > >
> > > > > We thank the reviewer for continuously engaging in the discussion.
> > > > >
> > > > > Whether or not to use statistical testing with reliable guarantees is indeed an ongoing discussion in the watermarking field, and we argue that to enable more trustworthy watermarks, using tests with statistical guarantees (i.e. reliable p-values) rather than a learned threshold is necessary.
> > > > >
> > > > > We agree with the reviewer that comparing our watermark with the original Unigram detector is important to ensure our results are comparable to the original work. Hence, we now compare in Figure 6 (Appendix B, page 20) the quality-detectability trade-off of our approach and the original implementation of the Unigram detector (using the method `dynamic_threshold` from the Unigram GitHub repo previously linked by the reviewer). We find that our detector implementation and the original Unigram detector yield very similar results.

---

### Author Response · Authors · 2025-11-19
**General Reply**

We have uploaded an updated version of the manuscript where the new content is marked in blue, and responded to all questions and concerns raised by the reviewers in individual responses below. When referencing the manuscript, we use the updated indexing (for section, figures, tables…). We are happy to further engage in the discussion in case of additional questions.

**GQ1: Why was Unigram not used as a baseline for our experiments?**

We explicitly did not include Unigram as a baseline for our work, as it has inherent statistical flaws that make it incomparable with most other watermarking methods. Importantly, Unigram does not control Type-1 error (i.e. the observed false positives can be *orders of magnitude* more than the expected one) making reported p-values unreliable. In particular, the Unigram scheme test assumes randomness with respect to the Red-Green split (Theorem 3.3 in [1]) yet the split is fixed and the same for all tokens. As we show empirically in Appendix B.2, the observed FPR can be 17 times higher than the expected one, which (as we show) is not the case for schemes with hashing. Importantly, this issue does not exist for us in isolation but has been previously observed in prior work [2] (in their Appendix E.2) and is more broadly referred to as key collision [3,4].

While Unigram may be used in specific settings (as in [2]) with explicit consideration of its Type-1 error in mind, we firmly believe that its inclusion as a general watermarking baseline is scientifically problematic and overall harmful to the field of watermarking. Prompted by the reviewer, we nonetheless moved the comparison to Unigram/PatternMark to the main body of our paper (Section 4.4) and added this important discussion (with the aforementioned experiments) to Appendix B.2. We are happy to engage further in the discussion if there is interest.

Orthogonally, regarding the security aspect of Unigram, [5] shows that the Red/Green split can be almost perfectly recovered by an adversary, leading to perfect watermark scrubbing and spoofing. On the contrary, attacks such as [6], also applicable to KGW schemes with context size 1 or 2, are less effective and more costly.

**References**

[1] Provable Robust Watermarking for AI-Generated Text, Zhao et al., ICLR 2024\
[2] Watermarking Autoregressive Image Generation, Jovanović et al., NeurIPS 2025\
[3] Distortion-free Watermarks are not Truly Distortion-free under Watermark Key Collisions, Wu et al., CoRR 2024\
[4] Pseudo- vs. True-Randomness: Rethinking Distortion-Free Watermarks of Language Models under Watermark Key Collisions, Zhang et al., CoRR 2024\
[5] Large Language Model Watermark Stealing With Mixed Integer Programming, Zhang et al., CoRR 2024\
[6] Watermark Stealing in Large Language Models, Jovanović et al., IMCL 2024

---

### Author Response · Authors · 2025-12-02
**Discussion Summary**

We thank the reviewers for their valuable feedback and for actively engaging in the discussion. We believe this rebuttal discussion has contributed significantly to improving our work. All reviewers acknowledged our rebuttal prior to the premature end of the discussion, which led to a final score of **(8, 8, 6, 6)**.

Reviewers highlighted the novelty of our work as the first watermark for diffusion language models (*d3D1, omL7, mAXZ*), as well as the novelty of our approach of designing a watermark through the scope of an optimization problem (*omL7, 3nzz*). They appreciate that the resulting watermarking algorithm is clear and intuitive (*d3D1, omL7*) and theoretically well grounded (*omL7, 3nzz*). At the same time, they also appreciate our comprehensive experimental evaluation (*3nzz, mAXZ*). The remaining concerns have been addressed in our rebuttal, as acknowledged by the reviewers. We summarize the discussion below (all changes in the manuscript are highlighted in blue):

- We now experimentally justify why we generally do not use Unigram/Patternmark as a baseline (*GQ1, d3D1 Q1, omL7 Q1–2, 3nzz Q1*): it is because in most settings Unigram is unreliable and lacks security. We have also moved the comparison of our approach with Unigram/Patternmark from the Appendix to the main body of our work. This experiment shows that our scheme outperforms Unigram/Patternmark, especially in the low-distortion regime, which is particularly relevant for the industry (*omL7 Q4*).
- We have updated our watermark implementation and show that our scheme’s runtime is actually similar to that of our baselines (*d3D1 Q3–4, omL7 Q3, mAXZ Q1, 3nzz Q2*). Also, as explained during the discussion, the theoretical complexity of our scheme is similar to that of popular AR watermarks such as SynthID-text (*d3D1 Q2*).
- We now compare our approach with an open-weight watermark, Gaussmark, and show that our approach significantly outperforms it (*mAXZ Q2*).
- Thanks to the exchange with reviewers (*omL7 Q5*), we have now expanded our theoretical derivations to self-hashing hash functions and now show that our watermark also yields a strong quality-detectability trade-off in these cases.
- We expanded our quality evaluation to include benchmark accuracies (*mAXZ Q3 + discussion*). We note that, with specific hyperparameter combinations and on some benchmarks, our watermark may induce a non-negligible drop in accuracy.
- We clarify parts of our work: the epsilon-parameterization algorithm (*d3D1 Q5*), our hash implementation (*omL7 Q6, 3nzz Q5*), the derivation for the expectation boost/predictive bias decomposition (*3nzz Q4*), and the fixed-point algorithm (*3nzz Q7–8*).
- We detail some choices of hyperparameters: we use random remasking as the default because it is the most adversarial setting for watermarking (*3nzz Q3*), and we use a KL constraint when designing the watermark because it leads to a closed-form solution and an efficient algorithm (*3nzz Q6*).
- We expand on our implementation of the Unigram detector, proving that it is more theoretically grounded than the original detector and we now experimentally validated that it yields similar performance (*d3D1 discussion*).
- We explain that our scheme is an extension of Red-Green watermarks that can be used with any token score distribution (*3nzz Q9*).

We thank the reviewers and the ACs for their efforts and refer to the individual rebuttals below for more details.

---

### Meta-Review · Area_Chair_5BVt · 2025-12-17

**Summary:**

In the discussion phases, all three reviewers who were negative about the paper are willing to raise their score. Another reviewer champions the paper with a score of 8. Given that reviewers are happy about the rebuttal, AC would recommend accept (poster) for this paper.

**Reviewer Concerns:**

Reviewer concerns AC believes are still outstanding:

1. From Reviewer d3D1: Implement Unigram faithfully, following their original formulations, rather than modifying them.

2. Reviewer mAXZ disagrees with the authors that benchmarks are not useful measures of quality for watermarking. The authors should be upfront in mentioning that their proposed scheme does appear to hurt generation quality to some nontrivial extent.

Except the above two concerns, all other concerns have been well addressed by the rebuttal, so that 3 reviewers are willing to raise their scores from negative to positive.

**Reviewer Scores:**

In the first round of review, Reviewer omL7, Reviewer mAXZ, and Reviewer 3nzz all scored this paper with 4 (marginally reject). However, after rebuttal, they mentioned that they are willing to raise the score. Another reviewer champions the paper with a score of 8. Therefore, the scores after the rebuttal will be 8666 or higher. AC would follow the reviewers' suggestion and recommend accept (poster) for this paper.

---

### Decision · Program_Chairs · 2026-01-26

Accept (Poster)